# MTCH2-mediated mitochondrial fusion drives exit from naïve pluripotency in embryonic stem cells

Amir Bahat[1], Andres Goldman[1], Yehudit Zaltsman[1], Dilshad H. Khan[2], Coral Halperin[1], Emmanuel Amzallag[1], Vladislav Krupalnik[3], Michael Mullokandov[1], Alon Silberman[1], Ayelet Erez[1], Aaron D. Schimmer[2], Jacob H. Hanna [iD] [3] & Atan Gross[1]

The role of mitochondria dynamics and its molecular regulators remains largely unknown during naïve-to-primed pluripotent cell interconversion. Here we report that mitochondrial MTCH2 is a regulator of mitochondrial fusion, essential for the naïve-to-primed interconversion of murine embryonic stem cells (ESCs). During this interconversion, wild-type ESCs elongate their mitochondria and slightly alter their glutamine utilization. In contrast, $MTCH2^{-/-}$ ESCs fail to elongate their mitochondria and to alter their metabolism, maintaining high levels of histone acetylation and expression of naïve pluripotency markers. Importantly, enforced mitochondria elongation by the pro-fusion protein Mitofusin (MFN) 2 or by a dominant negative form of the pro-fission protein dynamin-related protein (DRP) 1 is sufficient to drive the exit from naïve pluripotency of both $MTCH2^{-/-}$ and wild-type ESCs. Taken together, our data indicate that mitochondria elongation, governed by MTCH2, plays a critical role and constitutes an early driving force in the naïve-to-primed pluripotency interconversion of murine ESCs.

---

[1] Department of Biological Regulation, Weizmann Institute of Science, 7610001 Rehovot, Israel. [2] Princess Margaret Cancer Centre, University Health Network, Toronto, ON, Canada. [3] Department of Molecular Genetics, Weizmann Institute of Science, 7610001 Rehovot, Israel. These authors contributed equally: Amir Bahat, Andres Goldman. Correspondence and requests for materials should be addressed to A.G. (email: atan.gross@weizmann.ac.il)

Pluripotency pertains to the potential to differentiate into all three embryonic germ layers[1]. In vitro naïve pluripotency refers to embryonic stem cells (ESCs) derived from the inner cell mass of pre-implantation mouse blastocysts at E3.5, whereas in vitro primed pluripotency is represented by epiblast stem cells (EpiSCs) commonly derived from the late epiblast layer of post-implantation mouse embryos at E5.5–E6.5. These two cell types differ in their morphology, cytokine dependence, gene expression and epigenetic and metabolic profiles[2,3].

Reprogramming of mitochondria metabolism plays a critical role in the naïve-to-primed interconversion[4–12]. Importantly, this interconversion is accompanied by the elongation of the mitochondria and a pronounced metabolic switch to a highly glycolytic state with a low mitochondrial respiratory capacity[11], suggesting a role for mitochondria that is distinct from energy production. Mitochondria dynamics, which represents a central mechanism for cell identity and fate decisions[13–17], could potentially contribute to mitochondria enlargement and naïve-to-primed interconversion of ESCs. The critical cellular roles of mitochondrial dynamics during the early stages of embryogenesis are underscored by the observation that loss of mitochondria fusion results in embryonic lethality in midgestation[18,19].

Mitochondrial carrier homolog 2 (MTCH2) is a regulator of mitochondrial apoptosis and metabolism that plays a critical role in controlling the quiescence/cycling of hematopoietic stem cells (HSCs)[20–22]. We previously demonstrated that loss of MTCH2 results in embryonic lethality that is completed by E7.5[20]. Intriguingly, loss of MTCH2 also results in changes in mitochondria morphology[21–23], suggesting that MTCH2 may play a role in regulating mitochondrial dynamics during embryogenesis. Here we show that mitochondria elongation, governed by MTCH2, plays a critical role and constitutes an early driving force in the naïve-to-primed pluripotency interconversion of murine ESCs.

## Results

**MTCH2 is a regulator of mitochondrial fusion**. Our initial studies of the potential role of MTCH2 in mitochondrial dynamics were performed in mouse embryonic fibroblasts (MEFs), in which changes in mitochondria morphology are readily visualized. These studies demonstrated that knocking out MTCH2 in MEFs using Cre recombinase in vitro results in a less-elongated/round mitochondria morphology (Fig. 1a, top left and right panels, respectively), which was quantified by morphological classification (bottom left panel), aspect ratio calculation (bottom right panel), and sphericity (three-dimensional (3D) structural analysis of mitochondria using computerized segmentation; Fig. 1b). Importantly, the less-elongated/fragmented mitochondria morphology in the MTCH2[−/−] cells was rescued by the expression of MTCH2-GFP (Fig. 1c; 84.6% rescue of cells with elongated mitochondrial network as compared to MTCH2[F/F]). Next, we performed a mitochondria fusion assay using a photo-activatable GFP protein targeted to the mitochondria matrix[24], and found that MTCH2[−/−] mitochondria exhibit a lower fusion rate (slower spreading of the GFP; Fig. 1d). To confirm that increased mitochondrial fragmentation is indeed caused by a decrease in mitochondrial fusion, we expressed MFN2, a critical regulator of mitochondria fusion, in MTCH2[−/−] MEFs. Similar to the case of re-expression of MTCH2, MFN2-Myc decreased the mitochondria fragmentation (Fig. 1e; 86.1% rescue of cells with elongated mitochondrial network as compared to MTCH2[F/F]). Importantly, loss of MFN2, which results in complete fragmentation of mitochondria[18], was largely rescued by the expression of MTCH2-GFP (Fig. 1f), a finding that is consistent with the idea that MTCH2 is an important regulator of mitochondria fusion.

**MTCH2 loss in ESCs results in mitochondrial fragmentation**. Since MTCH2 knockout results in embryonic lethality by E7.5[20], we next asked whether this lethality is possibly due to a mitochondrial defect already present at E3.5, which could be studied in ESCs. We initially crossed MTCH2[+/−] mice and found that blastocysts grown in 2i (MEKi and GSK3i)/LIF (2i/L) yielded MTCH2[−/−] ESCs at a ratio lower than expected (Fig. 2a). Also, knockout of MTCH2 in MTCH2[F/F] naïve ESCs using Cre recombinase in vitro led to slower cell growth (Fig. 2b) with no effect on the basal levels of apoptosis (Supplementary Fig. 1a). Furthermore, MTCH2[−/−] ESCs preserved their naïve pluripotency, as evident from normal transcription levels of naïve core pluripotency markers (Supplementary Fig. 1b). We also tested the growth and differentiation capacity of these cells by assessing their ability to generate teratomas in vivo. Indeed, while the tumors obtained from the MTCH2[−/−] ESCs were smaller in size (Fig. 2c), they were able to differentiate properly to all three germ layers (Supplementary Fig. 1C). These findings indicate that MTCH2 deletion in ESCs does not compromise naïve pluripotency.

Next, we analyzed the morphology of mitochondria and found, similar to our findings in MEFs, that loss of MTCH2 results in a less-elongated/round mitochondria morphology (Fig. 2d), which was quantified by sphericity (Fig. 2e) and by aspect ratio (Fig. 2f). We also performed a mitochondria fusion assay, using ESCs prepared from genetically engineered mice with photo-activatable/convertible mitochondria (PhAM; green-to-red)[25] crossed to MTCH2[F/F] mice, and confirmed that MTCH2[−/−] mitochondria exhibit a lower fusion rate (slower decrease in red fluorescence protein (RFP) intensity; Supplementary Fig. 2A).

In an attempt to obtain insights into the molecular basis of the observed mitochondrial morphology changes in MTCH2[−/−] ESCs, we monitored the expression levels of the major mitochondria dynamics regulators in MTCH2[F/F] and MTCH2[−/−] naïve ESCs. Western blot analysis of whole-cell lysates and of mitochondria/cytosol fractions indicated that there is no significant differences in the expression levels of most of these proteins between the MTCH2[F/F] and MTCH2[−/−] cells (Supplementary Fig. 2B, C). Notably, there was a decrease in MFN1 and OPA1 levels in all three MTCH2[−/−] clones, but this decrease was accompanied by a decrease in TOM20 levels, which may indicate a reduction in mitochondria mass.

Notably, previous studies have shown that defects in mitochondria fusion lead to hyper-fragmentation of mitochondria, which, in turn, results in destabilization of mitochondrial DNA (mtDNA) and a decrease in mitochondrial function[26]. Indeed, we found that loss of MTCH2 in ESCs leads to a reduction in both mtDNA copy number (Fig. 2g) and mitochondria respiration (Fig. 2h). The above results are consistent with the idea that MTCH2 deletion in ESCs results in mitochondria hyper-fragmentation and decreased mitochondrial function, which lead to slower growth of ESCs and to a decrease in E3.5 derivation efficiency.

We previously reported that mitochondria respiration/function increases upon MTCH2 deletion in HSCs and skeletal muscle cells[21,22]. We hypothesize that loss of MTCH2 in these cells initially results in a decrease in mitochondria respiration/function (as seen in ESCs), and the increase observed is a compensatory mechanism to correct for the defect and keep the cells alive. In the case of ESCs, we hypothesize that increasing mitochondria respiration/function may not be sufficient to enable normal embryonic development, and thus the mitochondrial defect is not corrected and the embryos eventually die at E7.5.

**MTCH2 loss delays exit from naïve pluripotency**. The mitochondria of wild-type primed EpiSCs shift towards a more elongated state than those of naïve ESCs[11], suggesting that

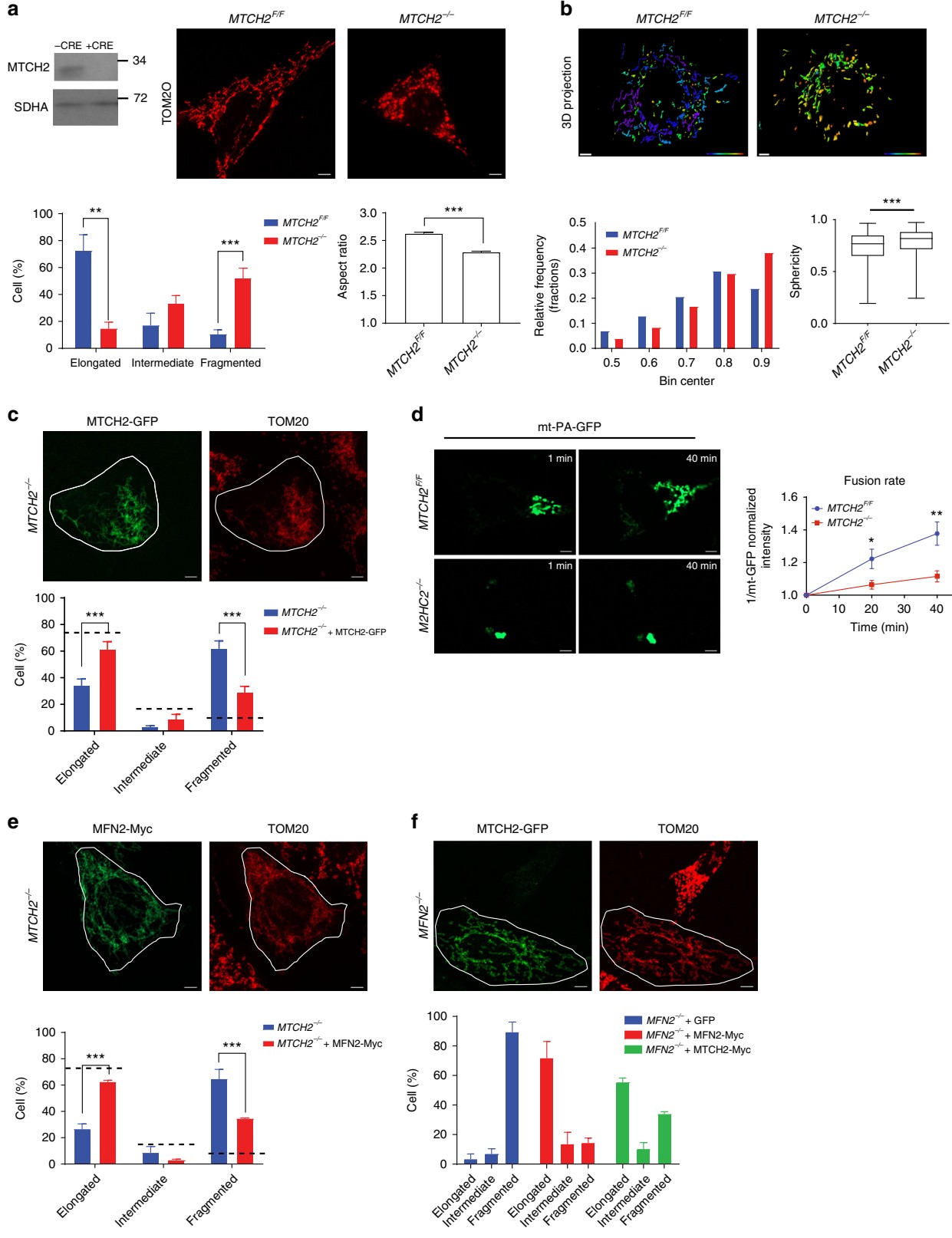

mitochondria fusion/elongation may play a role in the naïve-to-primed transition. Recent studies have demonstrated that in vitro priming for a period of 2 days of ground-state pluripotent ESCs (using Activin A/fibroblast growth factor 2 (FGF2); AF) generates pre-gastrulation epiblast-like cells (EpiLCs) that most closely resemble the in vivo counterpart E5.75 epiblasts[2]. Thus, to test our hypothesis that mitochondria fusion/elongation via MTCH2 is important for the naïve-to-primed interconversion, we utilized the ESCs-to-EpiLCs system. We monitored the morphology of mitochondria during the interconversion from ESCs to EpiLCs and found that the mitochondria of primed MTCH2^F/F cells were more elongated

**Fig. 1** MTCH2 is a regulator of mitochondrial fusion. **a** MTCH2 deletion results in mitochondria fragmentation. Top left panel: MTCH2 immunoblot of $MTCH2^{F/F}$ MEFs ± Cre recombinase in vitro. Succinate dehydrogenase A (SDHA) serves as loading control. Top right panels: representative immunofluorescence (IF) pictures of MEFs stained with anti-TOM20 antibody. Scale bar, 5 μm. Bottom left panel: quantification of cells according to their mitochondria network morphology ($n > 100$ cells scored; $n = 3$). Bottom right panel: aspect ratio calculation of mitochondria ($n > 60$ cells; $n = 3$). **b** MTCH2 deletion results in increased mitochondrial sphericity. Top panels: 3D reconstruction of MEFs mitochondria. Scale bar, 10 μm. Sphericity heat map, 0.404–0.972. Bottom left panel: histogram distribution of mitochondria sphericity. Bottom right panel: Box and whiskers plot of mitochondria sphericity. Results are presented as minimum, lower quartile, median, upper quartile, and maximum values ($n > 40$ cells). **c** MTCH2-GFP rescues $MTCH2^{-/-}$ MEFs fragmented mitochondria morphology. Top panels: Representative IF pictures of $MTCH2^{-/-}$ MEFs transfected with MTCH2-GFP and stained with anti-TOM20 antibody. Scale bar, 5 μm. Bottom panel: quantification of cells according to their mitochondrial network morphology. Dashed lines represent average mitochondrial network morphology of $MTCH2^{F/F}$ MEFs ($n > 100$ cells scored; $n = 3$). **d** $MTCH2^{-/-}$ mitochondria display lower fusion rate. Left and middle panels: live imaging pictures of MEFs 1 and 40 min after photoactivation, respectively. Right panel: quantification of mitochondrial fusion rate as 1/mt-GFP fluorescence intensity within the initially photoactivated region throughout time (30 cells were analyzed per group). Scale bar, 5 μm. **e** MFN2-Myc rescues the $MTCH2^{-/-}$ fragmented mitochondrial morphology. Representative IF pictures of MEFs transfected with MFN2-Myc (green) and stained with anti-TOM20 antibody (red). Scale bar, 5 μm. Bottom panel: quantification of cells according to their mitochondrial network morphology ($n > 60$ cells scored; $n = 3$). **f** MTCH2-GFP can partially rescue the $MFN2^{-/-}$-fragmented mitochondrial morphology. Representative IF pictures of MEFs transfected with MTCH2-GFP, and stained with anti-TOM20 antibody. Scale bar, 5 μm. Bottom panel: quantification of cells according to their mitochondrial network morphology ($n > 100$ cells scored; $n = 3$). Results in all graphs except for the ones in (**b**) are presented as mean ± SEM (*$p < 0.05$; **$p < 0.01$; and ***$p < 0.001$)

and also showed a steeper aspect-ratio increase as compared to the mitochondria of $MTCH2^{-/-}$ cells (Fig. 3a).

Since mitochondria dynamics represents a central mechanism for metabolic regulation[16,19], we also assessed whether metabolic changes accompany the naïve-to-primed interconversion. We measured the changes in metabolites' carbon labeling following 6 h of incubation with uniformly labeled $^{13}C$ substrates (as it was shown that isotopic equilibrium is reached by this time point in mouse ESCs[4]). Using gas-chromatography mass-spectrometry, we found that in $MTCH2^{F/F}$ EpiLCs there is a 10% decrease in relative glucose utilization and a corresponding increase in relative glutamine utilization for replenishing TCA-cycle metabolites (Fig. 3b, left and right panels, respectively). Notably, the labeling of citrate from glucose or directly from pyruvate only marginally changed after priming (Fig. 3b, left panel, and Supplementary Fig. 3A, respectively). Importantly, we found that the increase in relative glutamine utilization following priming of the $MTCH2^{F/F}$ cells is more pronounced than in the $MTCH2^{-/-}$ cells (Fig. 3c and Supplementary Fig. 3B), suggesting that MTCH2 plays a role in regulating this metabolic shift during the naïve-to-primed transition.

A previous study demonstrated that modulating the levels of acetyl-CoA and histone acetylation during the early differentiation of PSCs is essential for differentiation, pointing out that blocking histone deacetylation delays the exit from pluripotency[7]. To determine whether the inability of $MTCH2^{-/-}$ EpiLCs to properly alter their metabolism during priming correlates with histone acetylation levels, we assessed the changes in the levels of acetylated histone 3 (H3) by Western blot analysis. Interestingly, $MTCH2^{F/F}$ EpiLCs show robust H3 deacetylation while $MTCH2^{-/-}$ EpiLCs maintain high levels of H3 acetylation, comparable with the H3 acetylation levels seen in the naïve state (Fig. 3d). These results are consistent with the idea that MTCH2, via regulation of mitochondria morphology/metabolism, regulates epigenetic mechanisms. In an attempt to find a causal link between the metabolic results described above and the epigenetic results, we measured Acetyl-CoA (Ac-CoA) levels in whole-cell lysates but did not find any differences between the $MTCH2^{F/F}$ and $MTCH2^{-/-}$ naïve and primed cells (Supplementary Fig. 3C).

To assess whether the mitochondrial morphology/metabolism and epigenetic alterations in the $MTCH2^{-/-}$ EpiLCs also delay the exit from naïve pluripotency, we performed a genome-wide transcriptomic analysis with RNA-Seq. The analysis was performed on four groups of cells: $MTCH2^{F/F}$ cells grown in either 2i/L (ESCs) or in AF (EpiLCs) conditions, and $MTCH2^{-/-}$ cells grown in either 2i/L (ESCs) or in AF (EpiLCs) conditions. Hierarchical clustering and principal component analysis (PCA) based on differentially expressed (DE) genes between the four groups showed that while the $MTCH2^{F/F}$ ESCs and EpiLCs are distinct sub-populations of cells, $MTCH2^{-/-}$ ESCs and EpiLCs are clustered together and not with $MTCH2^{F/F}$ cells (Fig. 3e). These results suggest that priming has a dampening effect on the $MTCH2^{-/-}$ cells. Moreover, genes associated with either stem cell maintenance or stem cell differentiation (Fig. 3f, left and right panels, respectively) show large changes in AF versus 2i/L in $MTCH2^{F/F}$ cells (left columns in both panels) as opposed to relatively small or no changes in $MTCH2^{-/-}$ cells (right columns in both panels; note the many black lines that represent no difference).

To validate the RNA-Seq results, we utilized qPCR of known pluripotency and developmental genes regulated during priming. We found that $MTCH2^{-/-}$ EpiLCs failed to reduce the expression of naïve pluripotency genes and to adequately regulate the levels of developmental genes (Fig. 3g, top and lower panels, respectively). To confirm these changes at the protein level, we analyzed the levels of the naïve marker Nanog by immunostaining. While there was no difference in Nanog expression levels between $MTCH2^{F/F}$ and $MTCH2^{-/-}$ ESCs grown in 2i/L (Fig. 3h, top six pictures), its levels were extremely reduced in $MTCH2^{F/F}$ EpiLCs, whereas $MTCH2^{-/-}$ EpiLCs maintained relatively high expression levels of Nanog (Fig. 3h, bottom six pictures).

**Mitochondria elongation drives exit from naïve pluripotency.** To determine whether mitochondria fusion/elongation constitutes an early driving force in the naïve-to-primed pluripotency interconversion of ESCs, we tested whether rescuing the mitochondria's hyper-spherical/fragmented morphology in $MTCH2^{-/-}$ naïve ESCs by expressing MFN2-Myc would, in turn, restore exit from naïve pluripotency. Strikingly, $MTCH2^{-/-}$ naïve ESCs expressing MFN2-Myc showed repression of nuclear Nanog expression levels (Fig. 4a; see images in the left panels and bar graph in the top right panel). Importantly, in cells expressing MFN2-Myc there was a correlation between the increase in elongated mitochondria (increase in aspect ratio of mitochondria) and the decrease in Nanog expression levels (decrease in nuclear Nanog intensity; Fig. 4a, bottom right panel). We also determined the expression levels of estrogen receptor related β (ESRRβ), a downstream target of Naong and a key regulator of mouse naïve pluripotency[27], and found that

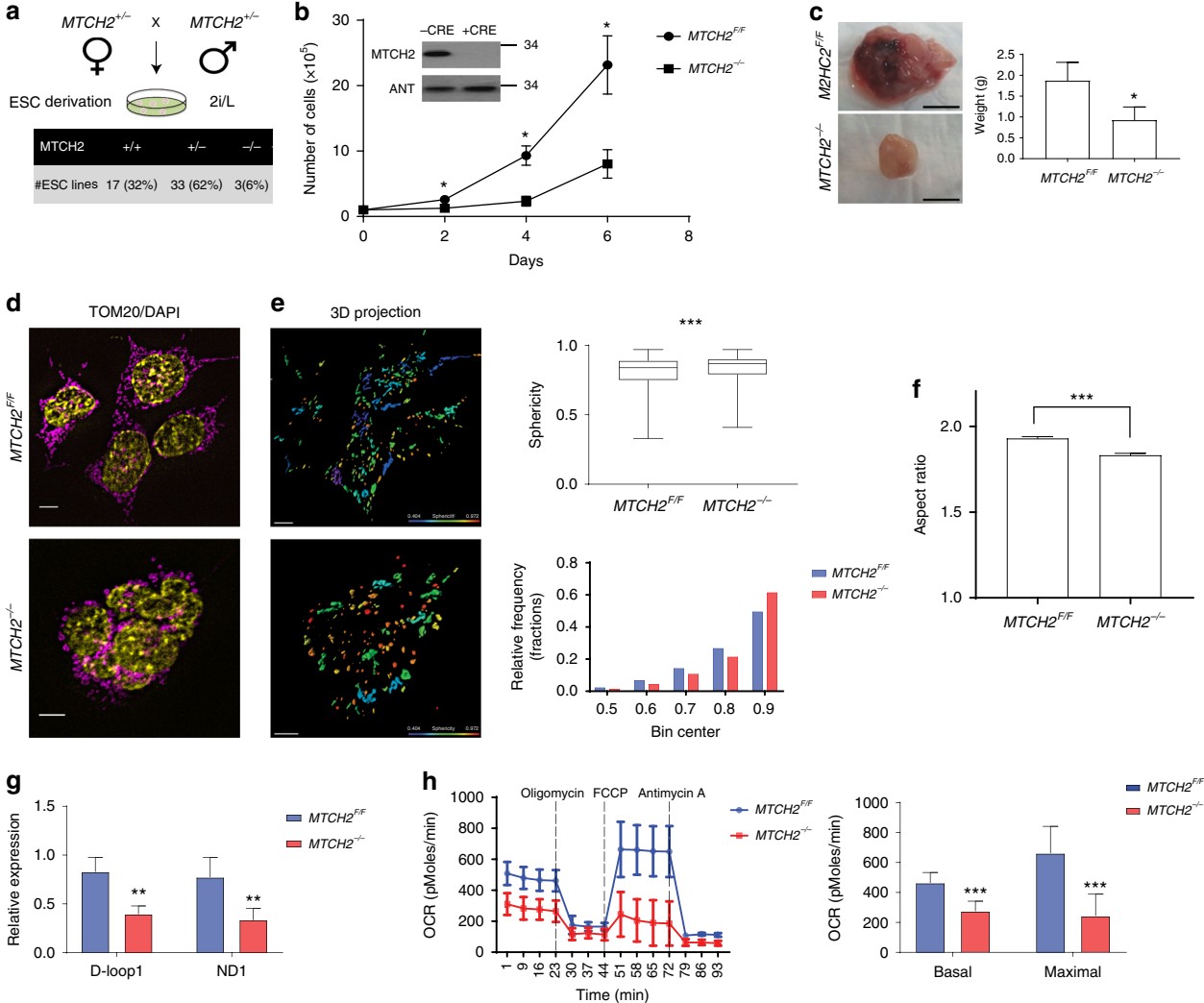

**Fig. 2** MTCH2 loss in ESCs results in mitochondrial fragmentation. **a** Blastocysts grown in 2i/L yielded $MTCH2^{-/-}$ ESCs at a ratio that is lower than the expected. Percentages of ESCs derivation efficiency after mating of MTCH2 heterozygous mice are presented. **b** Loss of MTCH2 in naïve ESCs results in slower cell growth. $MTCH2^{F/F}$ and $MTCH2^{-/-}$ ESCs growth curve. Results are presented as mean ± SD (*$p < 0.05$; $n = 4$). Inset: Immunoblot results of MTCH2 using anti-MTCH2 Abs of $MTCH2^{F/F}$ ESCs either left untreated (−CRE) or treated with Cre recombinant in vitro (+CRE). Mitochondrial adenine nucleotide translocator (ANT) serves as a loading control. **c** Teratomas obtained from the $MTCH2^{-/-}$ ESCs in vivo are smaller in size. Representative images are shown (scale bar, 1 cm). Results are presented as mean ± SEM (*$p < 0.05$; $n = 4$). Scale bar, 1 cm. **d** Loss of MTCH2 results in a less-elongated/round mitochondria morphology. Representative IF images (magenta, anti-TOM20 Abs; yellow, DAPI; Scale bar, 5 μM). **e** Loss of MTCH2 results in a hyper-spherical mitochondria morphology. Left panels: 3D mitochondria reconstruction of $MTCH2^{F/F}$ (upper panel) and $MTCH2^{-/-}$ (bottom panel) ESCs. Scale bar, 5 μM; Sphericity heat map, 0.404–0.972. Top right panel: Box and whiskers plot showing sphericity analysis of 3D reconstructed mitochondria. Results are presented as minimum, lower quartile, median, upper quartile, and maximum values (***$p < 0.001$; $n > 60$ cells were scored in three independent experiments)). Bottom right panel: Histogram representation of frequency distribution of the sphericity values of mitochondria. **f** Loss of MTCH2 resulted in a hyper-fragmented mitochondria morphology. Aspect ratio calculations of $MTCH2^{F/F}$ and $MTCH2^{-/-}$ mitochondria are presented. Results are presented as mean ± SEM (***$p < 0.001$; $n > 60$ cells were scored in three independent experiments. **g** Loss of MTCH2 in ESCs results in a decrease in mtDNA copy number. qPCR results of $MTCH2^{F/F}$ and $MTCH2^{-/-}$ ESCs mtDNA copy number are presented. Results are presented as mean ± SD (**$p < 0.01$; $n = 3$). **h** Loss of MTCH2 in ESCs results in a decrease in mitochondrial respiration. $MTCH2^{F/F}$ and $MTCH2^{-/-}$ ESCs were analyzed with the XF24 seahorse analyzer (left panel). Basal and maximal OCR/respiration are presented (right panel). Results are presented as mean ± SD (***$p < 0.001$; $n = 3$)

$MTCH2^{-/-}$ naïve ESCs expressing MFN2-Myc also showed repression of nuclear ESRRβ levels (Fig. 4b; see images in the left panels and bar graph in the top right panel). Importantly, in cells expressing MFN2-Myc there was a correlation between the increase in elongated mitochondria (increase in aspect ratio of mitochondria) and the decrease in ESRRβ expression levels (decrease in nuclear ESRRβ intensity; Fig. 4b, bottom right panel).

Most importantly, the expression of MFN2-Myc in $MTCH2^{F/F}$ naïve ESCs, maintained in naïve 2i/L conditions, also repressed nuclear Nanog expression levels (Fig. 4c). In the $MTCH2^{F/F}$ naïve ESCs expressing MFN2-Myc there was also a correlation between the increase in elongated mitochondria and the decrease in Nanog expression levels (Fig. 4c, bottom right panel). This correlation is further emphasized in cells expressing MFN2-Myc but whose mitochondria remained fragmented, in which case

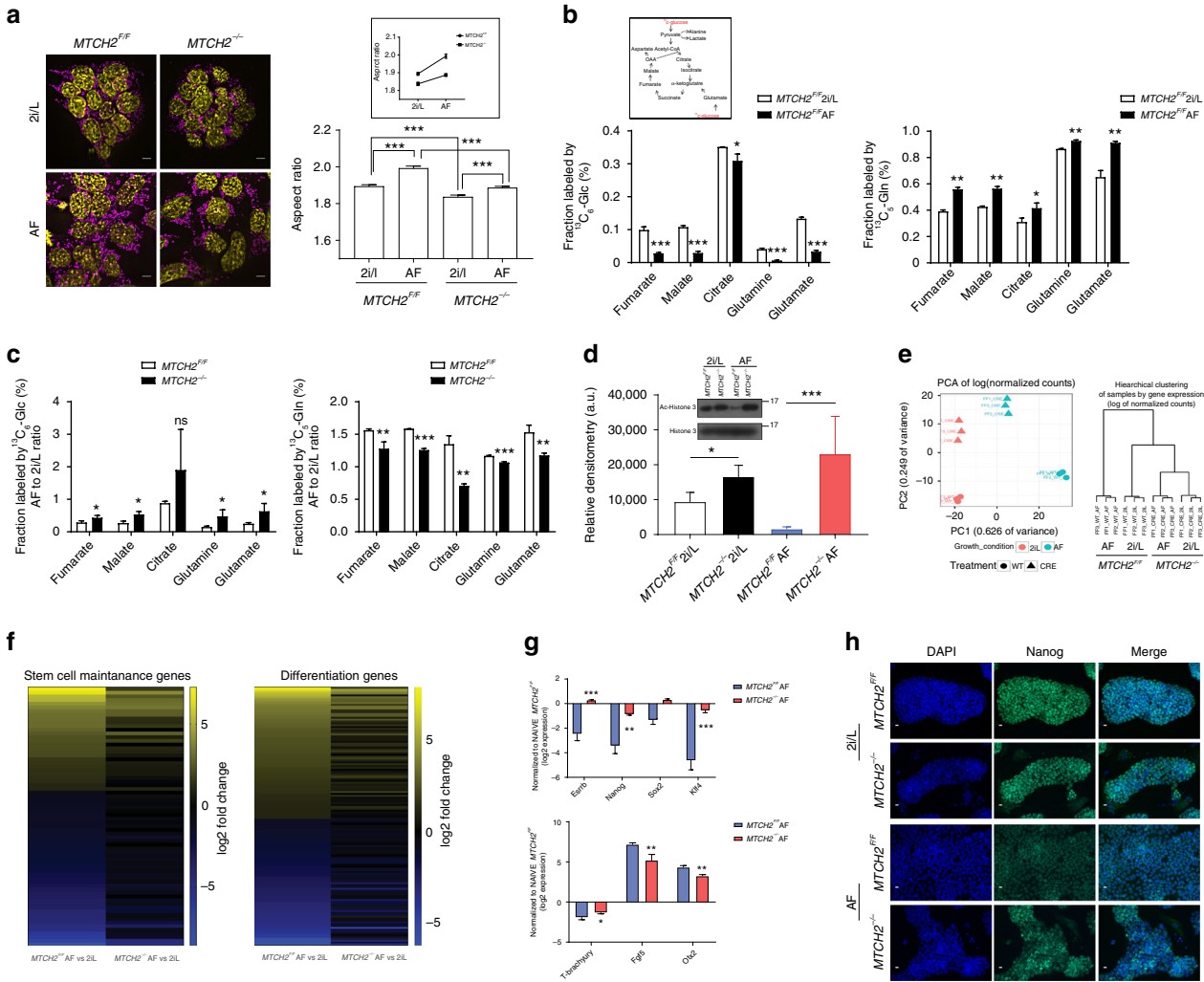

**Fig. 3 MTCH2 loss delays exit from naïve pluripotency. a** $MTCH2^{-/-}$ cells fail to adequately elongate mitochondria following priming. Left panels: Representative IF images (magenta, anti-TOM20 Abs; yellow, DAPI; Scale bar, 5 mM) of $MTCH2^{F/F}$ and $MTCH2^{-/-}$ cells. Right panel: Aspect ratio calculations of $MTCH2^{F/F}$ and $MTCH2^{-/-}$ cells mitochondria. Results are presented as mean ± SEM (***$p<0.001$; Anova statistical assay; n>60 cells were scored; n=3). Inset: Steeper elongation of $MTCH2^{F/F}$ mitochondria after priming. **b** $MTCH2^{F/F}$ EpiLCs preferentially increase glutamine utilization. Mass isotope tracing (Inset: schematic representation of TCA cycle) of $MTCH2^{F/F}$ ESCs and EpiLCs. Labeled fractions for each metabolite derived from either $^{13}C6$-glucose ($^{13}C$-glc; M+2 is shown; left panel) or from $^{13}C5$-glutamine ($^{13}C$-gln; M+4 or M+5 are shown; right panel) are plotted, after 6 hours incubation. Percentages were calculated using Metran software. Results are presented as mean ± SD (*$p<0.05$; **$p<0.01$, ***$p<0.001$; n=3). **c** $MTCH2^{-/-}$ cells fail to preferentially increase glutamine utilization following priming. Labeling and percentages were calculated as described in (B). Results are presented as mean ± SD (*$p≤0.05$; **$p<0.01$, ***$p<0.001$; n=3). **d** $MTCH2^{-/-}$ EpiLCs maintain high levels of acetylated histone 3. Inset: A representative Western blot is presented. Bottom panel: densitometry quantification of three separate Western blots. Results are presented as mean ± SD (*$p<0.05$; ***$p<0.001$; n=3). **e** $MTCH2^{-/-}$ ESCs and EpiLCs clustering similarities. Principal component analysis (PCA; left panel) and hierarchical clustering (right panel) of $MTCH2^{F/F}$ and $MTCH2^{-/-}$ cells based on differentially expressed genes (n=3). **f** $MTCH2^{-/-}$ cells show mild changes in gene expression following priming. Heat map showing the ratio of $MTCH2^{F/F}$ (left column) and of $MTCH2^{-/-}$ (right column) EpiLCs (AF) to ESCs (2i/L) RNA levels of stem cell maintenance genes (left panel) and stem cell differentiation genes (right panel) as obtained by RNA-seq (n=3). **g** $MTCH2^{-/-}$ EpiLCs fail to reduce the expression of naïve pluripotency genes. RNA levels of naïve pluripotency genes (top panel) and developmental genes (bottom panel) following priming. Results are presented as mean ± SD (*$p<0.05$; **$p<0.01$; ***$p<0.001$; n=3). **h** $MTCH2^{-/-}$ EpiLCs maintain high expression levels of Nanog. Representative IF images of $MTCH2^{F/F}$ and $MTCH2^{-/-}$ cells (n=3). (Scale bar, 10 μM)

Nanog expression levels were not decreased (see the cell marked with a dashed white line in the left panel images of Fig. 4c). Expression of MFN2-Myc in $MTCH2^{F/F}$ naïve ESCs, maintained in naïve 2i/L conditions, similarly repressed nuclear ESRRβ expression levels (Fig. 4d). In the $MTCH2^{F/F}$ naïve ESCs expressing MFN2-Myc there was also a correlation between the increase in elongated mitochondria and the decrease in ESRRβ expression levels (Fig. 4d, bottom right panel).

To assure that the changes in Nanog and ESRRβ expression levels were not due to a non-specific effect or to complete differentiation into a somatic lineage, we validated the expression levels of Oct-4, a pluripotent gene marker whose expression levels remain high during the naïve-to-primed interconversion[28]. We found that in $MTCH2^{F/F}$ naïve ESCs expressing MFN2-Myc and possessing elongated mitochondria, Oct-4 levels were only slightly reduced (Fig. 4e).

To distinguish the importance of active fusion per se from an elongated network for the exit from naïve pluripotency, we tested whether prompting the elongation of mitochondria by expressing a dominant negative form of the pro-fission protein dynamin-related protein 1 (DRP1-dn-YFP) would also induce exit from naïve pluripotency. We found that $MTCH2^{F/F}$ and $MTCH2^{-/-}$ naïve ESCs expressing DRP1-dn-YFP showed elongated mitochondria and repression of both Nanog and ESRRβ expression

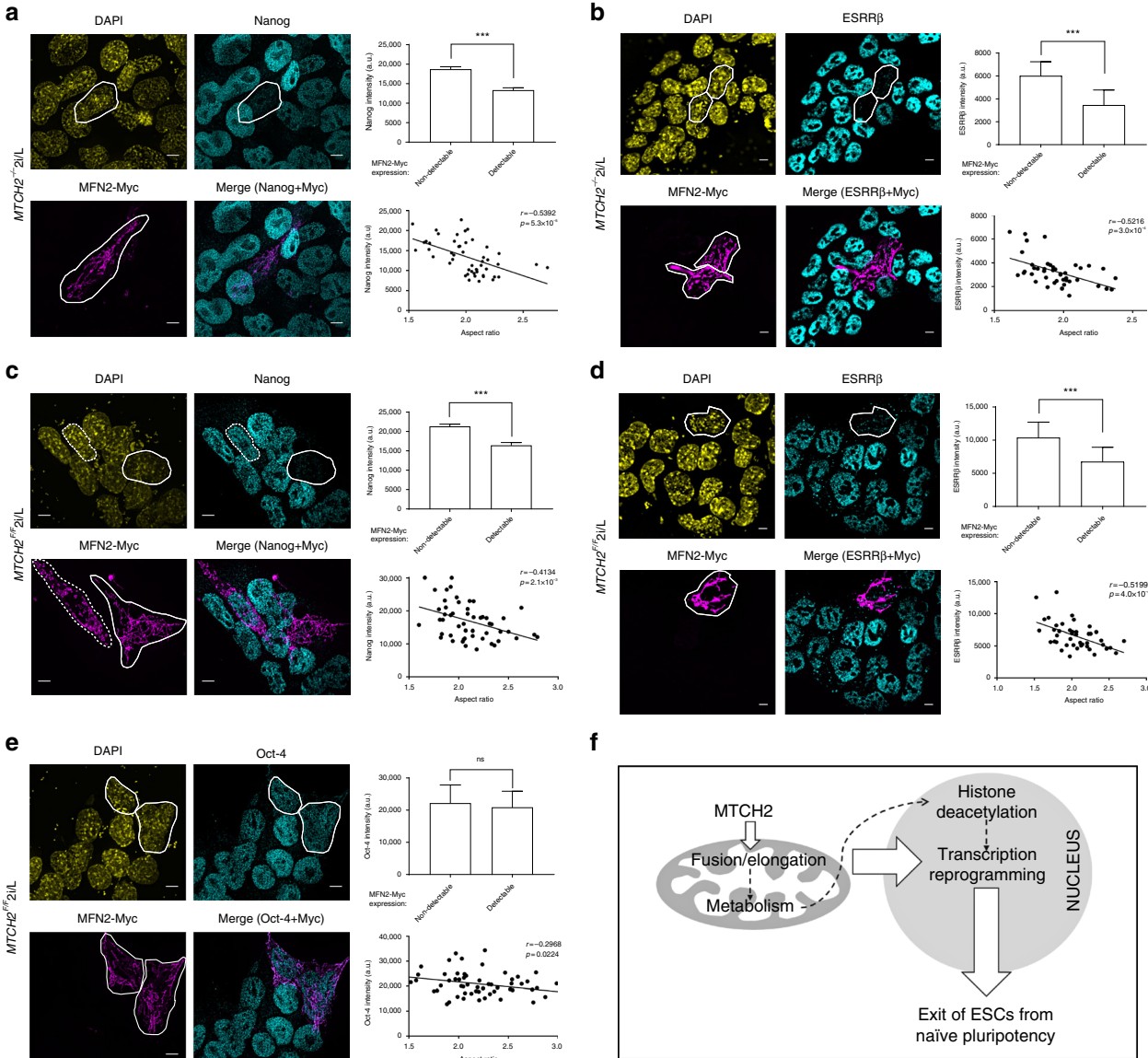

**Fig. 4** Mitochondria elongation drives exit from naïve pluripotency. **a** MFN2-Myc induces mitochondria fusion and represses Nanog expression in $MTCH2^{-/-}$ ESCs. Left panels: Representative IF images showing Nanog expression in $MTCH2^{-/-}$ ESCs expressing MFN2-Myc (scale bar, 5 μM). The cells expressing MFN2-Myc and repressing Nanog are marked with white lines. Top right panel: Average fluorescent intensity of Nanog in $MTCH2^{-/-}$ ESCs with either detectable or non-detectable expression of MFN2-Myc. Results are presented as mean ± SEM (***$p < 0.001$; $n > 50$ cells were scored for each group). Bottom right panel: $XY$ chart of average mitochondrial aspect ratio to mean intensity of Nanog levels in $MTCH2^{-/-}$ ESCs expressing MFN2-Myc ($n > 50$ cells were scored). **b** Inducing mitochondrial fusion by expressing MFN2-Myc in $MTCH2^{-/-}$ ESCs represses ESRRβ expression levels. Left panels: Representative IF images showing ESRRβ expression in $MTCH2^{-/-}$ ESCs expressing MFN2-Myc (scale bar, 5 μM). The cells expressing MFN2-Myc and repressing ESRRβ are marked with white lines. Top right panel: Average fluorescent intensity of ESRRβ in $MTCH2^{-/-}$ ESCs with either detectable or non-detectable expression of MFN2-Myc. Results are presented as mean ± SEM (***$p < 0.001$; $n > 50$ cells were scored for each group). Bottom right panel: $XY$ chart of average mitochondrial aspect ratio to mean intensity of ESRRβ levels in $MTCH2^{-/-}$ ESCs expressing MFN2-Myc ($n > 50$ cells were scored). **c** Expression of MFN2-Myc in $MTCH2^{F/F}$ ESCs induces mitochondria fusion and represses Nanog expression levels. Left panels: As in **a**. The cell marked with a dashed line, expresses MFN2-Myc yet possess fragmented mitochondria and maintains Nanog levels. Top and bottom right panels: As in (**a**). **d** Inducing mitochondrial fusion by expressing MFN2-Myc in $MTCH2^{F/F}$ ESCs represses ESRRβ expression levels. Left panels: As in (**b**). Top and bottom right panels: As in (**b**). **e** Inducing mitochondrial fusion in $MTCH2^{F/F}$ ESCs only slightly decreases Oct-4 expression levels. Left panels: Representative IF images showing Oct-4 expression in $MTCH2^{F/F}$ ESCs expressing MFN2-Myc (scale bar, 5 μM). The cells expressing MFN2-Myc are marked with white lines. Top and bottom right panels: As in (**a**). **f** Schematic representation of the role of mitochondrial fusion/elongation, governed by MTCH2, in regulating the exit from naïve pluripotency in ESCs

levels (Supplementary Fig. 4A–D; see images in the left panels and bar graphs in the right panels; note that DRP1-dn-YFP appears in green and is marked by white arrows in the top left images). Thus, active fusion per se is not required but rather an elongated network for the exit from naïve pluripotency.

## Discussion

Our studies reveal an unanticipated feature for mitochondria elongation in the naïve-to-primed interconversion of ESCs (Fig. 4f). We demonstrate that enforced elongation of mitochondria in naïve ESCs, even without eliminating the 2i/L conditions, is sufficient to trigger partial exit from naive pluripotency without entering differentiation, since the naïve markers Nanog and ESRRβ are repressed but Klf4 is not and the differentiation markers T-brachyury and FOXA2 are not expressed (Fig. 4 and Supplementary Fig. 5). These findings possibly relate to the recently described "Formative Pluripotency Phase", an intermediate phase proposed to exist as part of a developmental continuum between the naïve and primed pluripotent states, marked by the downregulation of only part of the essential transcription factor program that maintains naïve pluripotency[29,30]. Moreover, induction of lineage-specific markers was not evident in the formative phase, suggesting that terminating the naïve circuitry precedes the expression of lineage markers[31].

Our studies also show that MTCH2 is a direct regulator of mitochondrial fusion/elongation in both MEFs and ESCs, capable of rescuing the mitochondrial structure of MFN2-deficient cells. In ESCs, MTCH2 is important for altering glucose and glutamine utilization, histone deacetylation and nuclear gene reprogramming during the naïve-to-primed interconversion. However, further studies are required to elucidate the exact mechanism by which MTCH2 regulates mitochondria fusion/elongation and the metabolic-epigenetic link.

The finding that enforced elongation of mitochondria is sufficient for nuclear gene reprogramming is consistent with the idea that mitochondria elongation drives reprogramming of mitochondria metabolism, which, in turn, leads to histone deacetylation and changes in gene transcription (Fig. 4f; see dotted lines). Thus, mitochondria elongation, in which MTCH2 plays a critical role, is a necessary and early event important for naïve ESCs to adequately exit from the ground state of pluripotency.

## Methods

**Stem cell lines and cell culture**. Growth of primary murine pluripotent stem cells was conducted in FBS free N2B27-based media as described previously[32]. Briefly, 500 ml of N2B27 media was generated by including: 250 ml DMEM/F12 (HEPES free, custom made; Biological Industries), 250 ml Neurobasal (cat # 21103; Invitrogen), 2.5 ml N2 supplement (cat # 17502048; Invitrogen), 5 ml B27 supplement (cat # 17504044; Invitrogen), 1 mM glutamine (Invitrogen), 1% nonessential amino acids (Invitrogen), 0.1 mM β-mercaptoethanol (Sigma), penicillin–streptomycin (Invitrogen), 5 mg/ml BSA (Sigma). Naïve conditions for murine ESCs included 10 μg recombinant human LIF (L; Peprotech) and small-molecule inhibitors CHIR99021 (CH, 3 μM; Axon Medchem) and PD0325901 (PD, 1 μM; TOCRIS) termed 2i. Primed conditions for murine EpiLCs contained 8 ng/ml recombinant human bFGF (FGF2; Peprotech), 20 ng/ml recombinant human Activin A (Peprotech), and 1% knockout serum replacement (KSR; Invitrogen), in feeder-free growth factor reduced matrigel (Corning)-coated wells. For maintenance and Cre recombinase treatment, ESCs were kept in FBS/2i/L growth conditions, expanded in 500 ml of high-glucose DMEM (Invitrogen), 15% USDA certified fetal bovine serum (FBS; Biological Industries), 1 mM glutamine (Invitrogen), 1% nonessential amino acids (Invitrogen), 0.1 mM β-mercaptoethanol (Sigma), penicillinstreptomycin (Invitrogen), 10 μg recombinant human LIF (Peprotech) with 2i, on irradiation-inactivated mouse embryonic fibroblast (iMEF) feeder cells. Cells were maintained in 20% $O_2$ conditions and were passaged following 0.25% trypsinization (Invitrogen). Cell counts and growth curves were conducted using a hemacytometer (Sigma). MEFs were maintained in high glucose DMEM 10% FBS supplemented with glutamine, sodium pyruvate, and antibiotics.

**ESC derivation and MEF preparation**. The Weizmann Institute Animal Care and Use Committee approved all animal experiments. MTCH2 heterozygous or MTCH2 homozygous conditional knockout mice ($MTCH2^{F/F}$) (C57BL/6J strain;[20]) were crossed and E3.5 embryos were flushed and transferred onto iMEF-coated 96-well plates and cultured in FBS/2i/L conditions. Established ESC clones were genotyped by PCR and validated as MTCH2-deficient by qPCR and by Western blot analysis. $MTCH2^{F/F}$ mice were crossed to the PhAM mice[25] to generate $MTCH2^{F/F}$ PhAM mice carrying the mito-dendra2 transgene, and ESCs were prepared from these mice as described above. $MTCH2^{F/F}$ primary MEFs were prepared from 11 to 13-day-old embryos and transformed with the SV40 whole genome as described previously[33]. All the studies with MEFs described in the paper were performed with SV40-immortalized MEFs. Primer sequences used for genotyping are listed below:

MTCH2 WT F- 5′-TGTTCACAGGCTTGACTCCA-3′
MTCH2Δ F- 5′-TCCCAAGTGCTGGATTAAGG-3′
MTCH2 WT/Δ R- 5′- CAAACTGTATAGGTGAATGGCTCT-3′.

**In vitro naïve-to-primed pluripotent cell conversion**. For conversion of murine naïve pluripotent ESCs into primed EpiLCs, naïve ESCs were seeded in feeder-free N2B27 2i/L conditions for 16–24 h before switching to N2B27 Activin A/FGF2 (A/F) conditions for 2 days.

**Poly-A RNA sequencing**. Total RNA was extracted from the indicated cell cultures using PerfectPure RNA-cultured cell kit (cat # 2302340, 5′). To avoid DNA contaminations all samples were treated with DNase (5 prime). RNA integrity was evaluated on Bioanalyzer (Agilent 2100 Bioanalyzer), requiring a minimal RNA integrity number (RIN) of 8.5. Libraries were prepared according to Illumina's instructions accompanying the TruSeq RNA Sample Preparation Kit v2 (cat # RS-122–2001). Sequencing was carried out on Illumina HiSeq2500 according to the manufacturer's instructions, using 10 pM template per sample for cluster generation, and sequencing kit V2 (Illumina).

**RNA-Seq analysis**. Poly-A RNA sequencing was measured in four conditions: $MTCH2^{F/F}$ and $MTCH2^{-/-}$ in 2i/L or A/F. Each condition had three biological samples resulting in 12 RNA-seq samples. Sequencing libraries were prepared using the Illumina TruSeq mRNA kit. Reads were sequenced on an Illumina HiSeq 2500 SR60. Sequenced reads were mapped to the Mus Musculus genome version GRCm38 using TopHat v2.0.10[34]. Genes were identified using a.gtf obtained from ensembl release 82[35]. Per gene reads were counted using HTSeq[36]. Normalization of read counts and p-values for differential expression genes were computed using Deseq2[37]. Stem cell maintenance and developmental markers of DE genes were filtered using gene ontology consortium enrichment analysis.

**Histone isolation**. Histones were isolated by acid extraction as described previously[38] and were resolved (8.0–10 μg of histones) by 15% SDS–PAGE. Briefly, cells were suspended with NP-40 lysis buffer (10 mM Tris–Cl, [pH 7.6], 150 mM NaCl, 1.5 mM $MgCl_2$ 0.65% NP-40, 5 mM protease inhibitor) and homogenized, followed by 5000 r.p.m. centrifugation for 10 min at 4 °C. Cell pellets were resuspended in RSB buffer (10 mM Tris–Cl, [pH 7.4], 3 mM $MgCl_2$, 10 mM NaCl, 5 mM protease inhibitor) and homogenized, followed by the addition of 0.4 N $H_2SO_4$ for 1 h rotation in 4 °C. Samples were then centrifuged in 10,000 r.p.m. for 20 min at 4 °C and supernatant was added with 100% TCA to a final concentration of 18%. Samples were left on ice for 20 min, followed by centrifugation and washing the cell pellet once in acetone/HCl (5 μl acid/ml acetone) and twice solely in acetone followed by air-drying the pellets overnight and resuspention in water. Western blots were analyzed using anti-acetylated-H3 (cat # 06-599; Millipore) and anti-H3 (cat # 4499; Cell signaling) antibodies.

**Western blot and subcellular fractionation**. Subcellular fractionations were performed as described previously[20,39]. Briefly, cells were suspended in isotonic HIM buffer (200 mM mannitol, 70 mM sucrose, 1 mM EGTA, 10 mM HEPES (pH 7.5)). The supernatant was centrifuged at 10,000×g for 10 min to collect the mitochondrion-enriched fraction and the supernatant (cytosol). Proteins were size fractionated by SDS–PAGE and then transferred to polyvinylidene difluoride membranes. Antibodies used for Western blotting included anti-MTCH2 1:1000[20], anti-adenine nucleotide translocator 1:1000 (ANT; cat # SC-9300; Santa Cruz), and anti-succinate dehydrogenase A 1:1000 (SDHA; Ab 14715; Invitrogen), mouse anti-OPA1 1:1000 (cat # 612606; BD biosciences), mouse anti-MFN1 1:1000 (cat # NBP1–71775; Novus biologicals), mouse anti-MFN2 1:1000 (cat # WH0009927M3; Sigma-Aldrich), mouse anti-DRP1 1:1000 (cat # 611112; BD biosciences), rabbit anti-pDRP1 S637 1:1000 (cat # 4867; Cell signaling), anti-TOM20 1:1000 (cat # SC-11415; Santa cruz), and anti-actin 1:10,000 (cat # A5441; Sigma).

The uncropped scans of the blots presented in Fig. 3d appear in Supplementary Fig. 6.

**Quantitative PCR**. Total RNA was isolated using PerfectPure RNA cultured cell kit (5 prime), and gDNA was omitted by on-column DNase treatment. One

microgram of RNA was reverse-transcribed using HighCapacity cDNA Reverse Transcription Kit (Applied Biosystems). To evaluate mtDNA copy number, total DNA was isolated using Epicentre MasterPure DNA purification kit. To avoid RNA contaminations all samples were treated with RNase A (MasterPure). Quantitative PCR analysis was performed in triplicate for each sample by using 10 ng of the reverse transcription reaction in a Viia7 platform (Applied Biosystems) with Fast SYBR®Master Mix (Applied Biosystems). Data was extracted from the linear range of amplification. Error bars indicate standard deviation of triplicate measurements of three biological samples. Primer sequences used for all the genes tested are listed below:

MTCH2 F -5′-TGTTCACAGGCTTGACTCCA-3′′
MTCH2 R -5′-TGTTCACAGGCTTGACTCCA-3′
Actin F -5′-ATGAGCTGCCTGACGGCCAGGTCATC-3′
Actin R -5′-TGGTACCACCAGACAGCTCTGTG-3′
Gapdh F - 5′-TATGATGACATCAAGAAGGTGG-3′
Gapdh R -5′-CATTGTCATACCAGGAAATGAG-3′
Hprt F -5′-GCAGTACAGCCCCAAAATGG-3′
Hprt R -5′-GGTCCTTTTCACCAGCAAGCT-3′
D-loop1 F -5′-AATCTACCATCCTCCGTGAAACC-3′
D-loop1 R -5′-TCAGTTTAGCTACCCCCAAGTTTAA-3′
ND1 F -5′-TTACCAGAACTCTACTCAACT-3′
ND1 R -5′-ATCGTAACGGAAGCGTGGATA-3′
Esrrβ F -5′-CAGGCAAGGATGACAGACG-3′
Esrrβ R -5′-GAGACAGCACGAAGGACTGC-3′
Nanog F -5′-TCTTCCTGGTCCCCACAGTTT-3′
Nanog R -5′-GCAAGAATAGTTCTCGGGATGAA-3′
Sox2 F -5′-CATGAGAGCAAGTACTGGCAAG-3′
Sox2 R -5′-CCAACGATATCAACCTGCATGG-3′
Klf4 F -5′-TGGTGCTTGGTGAGTTGTGG-3′
Klf4 R -5′-GCTCCCCCGTTTGGTACCTT-3′
T Brachyury F -5′-GCTCTAAGGAACCACCGGTCATC-3′
T Brachyury R -5′-ATGGGACTCAGCATGGACAG-3′
Fgf5 F -5′-AATATTTGCTGTGTCTCAGG-3′
Fgf5 R -5′-TAAATTTGGCACTTGCATGG-3′
Otx2 F -5′-CATGATGTCTTATCTAAAGCAACCG-3′
Otx2 R -5′-GTCGAGCTGTGCCCTAGTA-3′.

**Teratoma formation assay.** For teratoma generation, $5 \times 10^6$ ESCs were injected subcutaneously into both flanks of recipient NSG immuno-deficient mice. Four weeks after initial injection tumors were harvested for paraffin embedding and sectioning followed by H&E staining.

**Respiration.** Measurement of intact cellular respiration was performed using the Seahorse XF24 analyzer (Seahorse Bioscience Inc.) and the XF Cell Mito Stress Test Kit according to manufacturer's instructions and as described[40]. Respiration was measured under basal conditions, and in response to Oligomycin (ATP synthase inhibitor; 0.5 μM) and the electron transport chain accelerator ionophore FCCP (trifluorocarbonylcyanide phenylhydrazone; 1 μM) to measure the maximal oxygen consumption rate (OCR). Finally, respiration was stopped by adding the electron transport chain inhibitor Antimycin A (1 μM).

**Cre recombinase treatment.** Recombinant His-TAT-NLS-Cre (Cre recombinase) fusion protein was expressed and purified as described previously[41]. Cre recombinase was diluted in DMEM/PBS to a final concentration of 3 μM, sterile-filtered, and incubated with the cells for 16 h. The cells were then washed with PBS, supplemented with growth medium, and grown for an additional 3–5 days before use in experiments.

**Expression of plasmids and transfections.** The MTCH2-GFP plasmid was generated in our laboratory and was previously described[39]. MFN2-Myc was purchased from Addgene (Plasmid #23213, David Chan). mito-pAGFP was kindly provided by Gyorgy Hajnoczky (Thomas Jefferson, USA). DRP1-dn-YFP was kindly provided by Andrew Gilmore (Manchester U, UK). Cells were transfected using the DNA-In Stem (ESCs) and DNA-In (MEFs) transfection reagents according to manufacturer's instructions (MTI; Global Systems).

**Imaging.** Confocal microscopy images were captured with a Carl zeiss lsm 710 confocal microscope equipped with yokogawa spinning disc using a ×100 1.4 NA Apochromat objective (ZEISS). The excitation wavelengths for DAPI (405), GFP, Cy2 (488), Cy3, and Mitotracker Red CMXRos (561; Molecular Probes, Inc.). Fluorescence images were acquired using cellSens software with Olympus IX83 (Olympus, Japan) equipped with ×60 oil immersion objective, Orca Flash4.0 camera (Hamamatsu Photonics, Japan) and a LED light source (CoolLED, UK).

**Immunofluorescence.** Cells were grown in 13 mm sterile coversilps, washed with PBS and fixed with 4% PFA for 15 min at 37 °C. PFA was quenched by incubating the cells for 10 min in 50 mM NH4Cl2 in PBS. After washing, cells were permeabilized with 0.1% triton x-100 for 20 min and then blocked for 1 h at room

temperature (RT) with 10% BSA 0.1% triton x-100 in PBS. Cells were probed with mouse anti-Myc 1:200 (cat # SC-40; Santa cruz), rabbit anti-Myc 1:200 (cat # CST2278; Cell signaling technology), rabbit anti-TOM20 1:200 (cat # SC-11415; Santa cruz), mouse anti-ATP synthase (Complex-V-β) 1:200 (cat # MS-503; Mitosciences), rabbit anti-Nanog 1:200 (cat # A300–397A; Bethyl), mouse anti-Oct3/4 1:200 (cat # SC-5279; Santa cruz), mouse anti-ESRRβ 1:200 (cat # H6705; R&D systems), goat anti-T brachyury 1:200 (cat # SC-17743; Santa cruz), and goat anti-FOXA2/HNF-3β 1:200 (cat # SC-6544; Santa cruz) Abs. Next, cells were washed with PBS, stained with second Ab conjugated to Cy2 or Cy3 dyes (Molecular Probes, Inc.), washed, mounted with SlowFade Light Antifade kit (Molecular Probes, Inc.), and analyzed by confocal microscopy.

**Aspect ratio of mitochondria calculations.** Aspect ratio (the ratio between the major and minor axis of the ellipse) was calculated using FIJI software. Z-stacks of cells stained for the mitochondrial marker TOM20 were processed and thresholded (binarized) to run the analyzed particles algorithm and calculate aspect ratio of each mitochondria.

**Analysis of Nanog, ESRRβ, and Oct-4 nuclear expression levels in images.** Analysis of Nanog, ESRRβ, and Oct-4 mean fluorescence intensity was performed on Z projections of z-stack pictures of ESCs transfected with MFN2-Myc. Nuclei perimeters were drawn using the FIJI poligon tool, and mean fluorescent intensity was measured in arbitrary units for each of the different regions of interest. Average fluorescence intensity was calculated for cells expressing MFN2-Myc and cells with no detectable expression of the protein. To calculate the correlation between aspect ratio of mitochondria and nuclear Nanog, ESRRβ or nuclear Oct-4 fluorescence intensities, each MFN2-Myc-expressing cell was analyzed separately. Using the fluorescence intensity of the MFN2-Myc channel we performed aspect ratio analysis of mitochondria as described above. Then the mean nuclear fluorescence intensity of either Nanog or Oct-4 in the MFN2-Myc-expressing cell was measured and analyzed together with its average aspect ratio value.

**Mitochondria 3D reconstruction and morphometric analysis.** Cells were fixed and mitochondria was stained as described above. All the images were taken under the same conditions of laser power and exposure time. Z-stack interval was set to 0.24 μm and a 3D deconvolution algorithm was applied to the ESCs images using Autoquant software (MediaCybernetics). Mitochondrial 3D reconstruction was achieved using segmentation algorithm of Imaris software (Bitplane).

**Quantification of mitochondrial fusion rate in MEF.** For quantification of mitochondrial fusion rate in MEF, $2 \times 10^5$ cells seeded onto 24-mm round glass plates were transfected with mito-pAGFP. After 24 h, cells were placed on the stage of a spinning disc microscope coupled to a FRAP unit (Zeiss 710). To activate the PA-GFP fluorescence, 1z plane was activated using 100% of the power of the 405 nm laser line with a ×100, 1.4 NA objective. Frames were then acquired every 20 min using the 488 nm and the 563 laser. Intensity of the green fluorescence was measured in the initially photoactivated ROI through time. Mitochondria were stained using 50 nM Mitotracker CMXRos (Molecular Probes, Inc.).

**Quantification of mitochondrial fusion rate in ESCs prepared from MTCH2F/F PhAM mice.** For these experiments we used genetically engineered mice with photo-activatable mitochondria (PhAM)[25]. These mice express mito-dendra2, a mitochondria matrix-targeted green-to-red photoconvertible fluorescent protein, in which mitochondria are constantly green, and laser excitation converts the excised population of mitochondria to become red. PhAM mice were crossed to $MTCH2^{F/F}$ mice to generate $MTCH2^{F/F}$ PhAM mice, and quantification of mitochondrial fusion rate was analyzed by fluorescence spreading in ESCs prepared from these mice. Mito-dendra2 was exposed to 405 nm laser light irreversibly switching green fluorescent mitochondria to red fluorescent mitochondria. Images were acquired every 30 min with a Zeiss Yokogawa spinning disc scanning unit coupled with an inverted IX83 microscope (Olympus), using a ×100 oil lens (NA 1.4) and capturing the images with sCMOS 4.2 Mpixel camera controlled by VisiView software (GFP [488 nm], RFP [561 nm]). Images were analyzed using Fiji software.

**Metabolomics analysis.** End-point analysis of metabolites using gas chromatography–mass spectrometry (GC–MS) was performed as previously described[42]. Briefly, ESCs were incubated with either 10 mM of D-glucose (cat # CLM-1396; U-$^{13}$C$_6$, 98%, Cambridge Isotope Laboratories) or 4 mM of L-glutamine (cat # CLM-1822; U-$^{13}$C$_5$, 98%, Cambridge Isotope Laboratories) or 10 mM of sodium pyruvate (cat # CLM-2440; U-$^{13}$C$_3$, 98%, Cambridge Isotope Laboratories) for 6 h. Subsequently, cells were washed with ice-cold saline, lysed with 50% methanol in water and quickly scraped followed by three freeze–thaw cycles in liquid nitrogen. The insoluble material was pelleted in a cooled centrifuge (4 °C) and the supernatant was collected for consequent GC–MS analysis. Samples were dried under air flow at 42 °C using a Techne Dry-Block Heater with sample concentrator (Bibby Scientific) and the dried samples were treated with 40 μl of a methoxyamine hydrochloride solution (20 mg/ml in pyridine) at 37 °C for 90 min

while shaking followed by incubation with 70 µl N,O-bis (trimethylsilyl) tri-fluoroacetamide (Sigma) at 37 °C for an additional 30 min.

**Acetyl-CoA sample preparation and LC–MS/MS analysis**. Acetyl-CoA was extracted as described previously[43]. Briefly, cold methanol (500 µl; −20 °C) was added to the cell pellets, $^{13}C_2$-acetyl-CoA (5 µl of 500 ng/ml aqueous solution) was added as internal standard (Sigma-Aldrich), and the mixture was shaken for 30 s (10 °C, 2000 r.p.m., Thermomixer C, Eppendorf). Cold chloroform (500 µl; −20 °C) was added, the mixture was shaken for another 30 s, and then 200 µl of water (4 °C) was added. After the mixture was shaken for 30 s and left on ice for 10 min, it was centrifuged (21,000×$g$, 4 °C, 10 min). The upper layer was collected and evaporated. The dry residue was re-dissolved in eluent buffer (500 µl) and centrifuged (21,000×$g$, 4 °C, 10 min) before placing in LC–MS vials.

Acetyl CoA was analyzed as described previously[44]. Briefly, the LC–MS/MS instrument consisted of an Acquity I-class UPLC system and Xevo TQ-S triple quadrupole mass spectrometer (both Waters) equipped with an electrospray ion source. LC was performed using a $100 \times 2.1$-mm i.d., 1.7-µm UPLC Kinetex XB-C18 column (Phenomenex) with mobile phases A (10 mM ammonium acetate and 5 mM ammonium hydrocarbonate buffer, pH 7.0, adjusted with 10% acetic acid) and B (acetonitrile) at a flow rate of 0.3 ml min$^{-1}$ and column temperature of 25 °C. The gradient was as follows: 0–5.5 min, linear increase 0–25% B, then 5.5–6.0 min, linear increase till 100% B, 6.0–7.0 min, hold at 100% B, 7.0–7.5 min, back to 0% B, and equilibration at 0% B for 2.5 min. Samples kept at 4 °C were automatically injected in a volume of 5 µl. Mass spectrometry was performed in positive ion mode, monitoring the MS/MS transitions $m/z$ 810.02 → 428.04 and 810.02 → 303.13 for acetyl-CoA and 812.03 → 428.04 and 810.02 → 305.13 for $^{13}C_2$-acetyl-CoA.

**Apoptosis analysis**. Cells were stained with Annexin V detection kit (BD Biosciences) according to the manufacturer's instructions, coupled with propidium iodide (PI; Sigma) staining and analyzed by flow cytometry.

**Statistical analysis**. $P$ values were calculated by using Student's $t$-test, unless otherwise specified, with GraphPad prism statistical software.

## Data availability
The authors declare that the main data supporting the findings of this study are available within the article and its Supplementary Information. Extra data are available from the corresponding author upon request. RNA-Seq analysis was deposited in National Center for Biotechnology Information (NCBI) at gene expression omnibus (GEO) under the accession number GSE122067.

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

## Acknowledgements

We are grateful to Alon Harmelin, Rebecca Haffner, Alina Maizenberg, Golda Damari, and Calanit Raanan for help with generating the *MTCH2^{F/F}* mice. We also thank David Chan (Caltech) for MFN2 expression plasmids and help with protocols and Ofra Golani for help with microscopy. We are grateful to all the members of the Gross Laboratory for their support, and discussion and comments on the manuscript. This study was supported in part by the Israel Science Foundation (ISF), USA–Israel Binational Science Foundation (BSF), German-Israel Foundation (GIF), Minerva Stiftung, and a Hyman T. Milgrom Trust grant. A.G. is the incumbent of the Marketa and Frederick Alexander Professorial Chair.

## Author contributions

A.B. and A.Goldman performed most of the experiments presented in the paper. Y.Z., C.H., E.A., V.K. and M.M. helped with many of the experiments described. D.H.K. and A.D.S. performed the histone acetylation analysis, V.K. and J.H.H. performed part of the stem cell experiments, and A.S. and A.E. helped with metabolomics analysis. A.B., A.Goldman, and A.Gross planned the projects and wrote the paper. All authors discussed the results and commented on the manuscript.

## Additional information

**Competing interests:** The authors declare no competing interests.

