## [Peer Review File · Nature Communications]

Reviewers' comments:

Reviewer #1 (Remarks to the Author):

MTCH2-mediated mitochondrial fusion drives exit from naïve pluripotency in embryonic stem cells

It has become clear that the glycolytic metabolism is distinct between different states of pluripotency in the recent years. The authors here identify a novel regulator for mitochondrial fusion, mitochondrial carrier homolog 2 (MTCH2), which is critical for this transition (naïve to primed) in mouse embryonic stem cells (mESCs). Cells lacking this marker failed to exit naïve pluripotency, while overexpression of the profusion marker mitofusin 2 (MFN2) rescued this phenotype. The authors build on their previous interesting work describing that a loss of mitochondrial dynamics results in embryonic lethality during development, suggesting that these embryos failed to properly switch their metabolic dynamics towards a more primed state of pluripotency. By using the naïve to primed interconversion the authors concurrently provide valuable insights on the role of mitochondrial metabolism during early development.

The authors apply a whole array of different techniques to support and validate their findings, adding to the comprehensive nature of their research and strengthening the scientific value of their conclusions.

The authors primarily validated the effects of MTCH2 knock-out (KO) in mouse embryonic fibroblasts (MEFs). They clearly show a rescue of MTCH^{-/-} with MTCH-GFP, however it would be interesting to see a comparison between MTCH^{-/-}+MTCH-GFP and MTCHF/F. They further show that expression of MFN2, a critical regulator of mitochondrial fusion, reduced mitochondrial fragmentation, which definitively proves that the cause of this fragmentation is a lower fusion rate. As with the MTCH-GFP comparison, a comparison with MTCHF/F may serve to further confirm their findings.

The authors use a series of elegant experiments to show that MTCH2 KO maintains mESCs in a naïve state of pluripotency. They firstly show that they could derive MTCH2^{-/-} ESCs in 2i LIF conditions, albeit at a lower derivation efficiency than expected. Furthermore, when comparing MTCH2^{F/F} to MTCH2 KO with Cre recombinase, the authors observe slower cell growth, however interestingly the MTCH2 KO cells preserved naïve pluripotency. It would add to the scientific strength of the paper if the authors could speculate on the cause of the observed reduction in derivation efficiency or growth. The authors were able to further couple the effects of mitochondrial fusion to previously reported destabilization of mitochondrial DNA and mitochondrial respiratory function. They show that MTCH2 regulates mitochondrial dynamics required for the exit from naïve pluripotency by comparing the priming of MTCH2^{F/F} cells and MTCH2^{-/-} towards EpiLCs. The authors observed a dampening in metabolic shift from pyruvate oxidation to a more primed metabolism, mitochondrial glutaminolysis. It is interesting that mitochondrial pyruvate oxidation is reduced in primed EpiLCs, while it has been previously shown that primed EpiSCs have a higher glycolytic rate compared to mESCs (Gu et al., Cell Stem Cell. 2016, Zhou et al. EMBO J. 2012), resulting in higher levels of pyruvate in the cytoplasm.

The authors were also able to further relate the inability of MTCH2^{-/-} EpiLCs to properly shift their metabolism to mitochondrial glutaminolysis to high levels of H3 acetylation. This successfully demonstrates not only that the cells exhibited hallmarks of naïve pluripotency, but also that MTCH2 is involved in the regulation of epigenetic mechanisms.

Their transcriptomic analysis serves as a thorough validation of the effect of MTCH2 on pluripotency, contributing a useful dataset to the stem cell community. The addition of qPCR and immunostaining constitutes an appropriate validation of the RNASeq data.

Finally, the authors adequately validate mitochondrial elongation as an early driving force of the naïve to primed pluripotency interconversion by correlating the effects of MFN2-Myc overexpression to mitochondrial morphology, NANOG and OCT4 expression levels. Taken together the authors effectively show that a shift in mitochondrial morphology is sufficient for reprogramming of mitochondrial metabolism, which in turn leads to epigenetic and transcriptomic changes consistent with the naïve to primed interconversion.

Overall sentence build could be improved, for greater clarity and readability. There are many spelling errors, even in the abstract. The authors should address these.

Lines 49-53, good background overview, but a bit cumbersome for the reader

Lines 121-123 repetitive

Line 136-138 needs to be clarified in more detail, to clearly show that there are two groups: MTCH2^{-/-} and MTCH2^{F/F}, which both have ESCs and EpiLCs. It was difficult to immediately understand this without referring to the figure.

Reviewer #2 (Remarks to the Author):

The manuscript from Atan Gross and colleagues describes the effects of genetic inactivation of MTCH2 in MEFs and pluripotent cells. MTCH2 regulates mitochondria fusion in MEFs, while its absence in ES cells leads to fragmentation and reduced biogenesis and respiration. During the initial 48 hours of exit from naïve pluripotency mitochondria start to elongate and shift their metabolism from Glucose to Glutamine consumption. All these effects are mitigated in the absence of MTCH2. Moreover, the transcriptional changes associated with exit from naïve pluripotency are in large part abrogated in MTCH2 null ES cells.

Experiments are well performed, statistical analyses seems appropriate, and results are in large part novel, while some confirm previous studies.

A few points should be addressed to strengthen authors' conclusions.

1) no direct measurement of fusion in ES cells has been performed. The authors should measure GFP spreading as in figure 1D. This would support the claim that MTCH2 directly regulates fusion in ES cells, as indicated in the title and abstract of the manuscript.

2) the flux analysis presented in figure 3B shows that in MATCH2 fl/fl after 48h in AF there is a drastic reduction in some metabolites (e.g. Fumarate, Malate and Glutamine) labelled by ¹³C-Glucose.

Panel C shows the AF to 2i/L ratio in MTCH2 fl/fl and ^{-/-} cells.

Somehow the two panels do not appear to be consistent, because in 3C the AF to 2i/L ratio in MATCH2 fl/fl is between 0.6 and 0.8 for Fumarate, Malate and Glutamine, indicating only a mild reduction in labelling of such metabolites.

To avoid confusion I would suggest to present both datasets in the same way, plotting the fraction of metabolites labelled by ¹³C-Glc/Gln, as in Figure 3B.

3) the results indicating increased levels of Histone Acetylation are interesting, yet there is not a clear causal link with the metabolic changes observed.

Is Acetyl-CoA more abundant in MTCH2^{-/-} ES cells?

4) In figure 4 the authors claim that expression of MFN2 is sufficient to drive exit from naive pluripotency. Such conclusions are based on the downregulation of Nanog. Given that Nanog appear dispensable for maintenance/induction of naive pluripotency, the authors should support their conclusions by looking at other naive markers (e.g. Klf4, Esrrb, Tfcp2l1) and early differentiation markers (e.g. Fgf5 and Otx2).

5) minor comment: line 80-81: rephrase, lethality by E7.5 cannot be due to defects in ESCs, rather due to defects already present at E3.5, which could be studied in ESCs.

6) minor comment: a previous work from the Gross laboratory characterised MTCH2 in ESCs as a regulator of apoptosis via tBID. Is such role completely unrelated from the effects observed in the current manuscript? The authors should discuss the two studies and the possible implications.

Reviewer #3 (Remarks to the Author):

The mitochondrial protein MTCH2 has been previously described to regulate mitochondrial functions as well as apoptosis and metabolism. MTCH2 was also found to be an important player in the regulation of hematopoietic stem cell fate by the authors of this manuscript. Here, they have revealed that MTCH2 is also required for naïve-to-primed interconversion of murine ESCs and have linked this to its novel role as a promoter of mitochondrial fusion. This is of relevance to the growing interest in mitochondrial dynamics and metabolism as key regulators of stem cell fate. Their characterization of MTCH2^{-/-} ESCs reveals interesting differences in mitochondrial metabolism between WT and MTCH2^{-/-}, particularly when studying the shift between naïve to primed states. Interestingly, overexpression of the mitochondrial fusion factor MFN2 appears to be sufficient to promote ESC reprogramming. However, the underlying mechanism is unclear and therefore the argument that normal regulation of mitochondrial metabolism and histone acetylation is downstream of MTCH2-dependent mitochondrial fusion remains largely correlative and the underlying mechanism remains to be defined.

Major points:

1. How does MTCH2 affect mitochondrial fusion? The data in Fig 1 convincingly shows that MTCH2 KO results in a fragmented morphology. The results of a PA-GFP fusion-experiment combined with the partial rescue of mitochondrial morphology in MFN2^{-/-} cells overexpressing MTCH2 suggests that it may play a pro-fusion role. It should be noted that the mitochondrial morphology changes are far less pronounced in ESC WT vs MTCH2^{-/-} cells. The authors previously showed in LSK cells (Maryanovich et al., 2015) that MTCH2 depletion causes mitochondrial enlargement due to reduced levels of mitochondrial Drp1. How can these discrepancies be explained? Drp1 phosphorylation status could be looked at along with a blot showing how all major mitochondrial dynamics factors are affected by MTCH2. Without further mechanistic insight it remains a possibility that mitochondrial dynamics regulation by MTCH2 is an indirect effect of metabolic changes. Similarly, how can the authors exclude that MTCH2 regulation of apoptosis is not playing a role at all in ESC cell pluripotency?

2. Another discrepancy exists in the regulation of OXPHOS by MTCH2. In murine hematopoietic stem cells and skeletal muscle, MTCH2 loss increases OXPHOS. This should be discussed. It is also unclear how significant the OXPHOS changes are with regards to the MTCH2^{-/-} ESC cell fate.

3. In Fig. 4, MFN2 overexpression is used to drive mitochondrial fusion and to argue that this helps

drive ESC naive-primed conversion. This is a key message of the manuscript and therefore other methods to substantiate these data are important. How certain can the authors be that alternative functions of MFN2 are not in play. For example, both MFN2 and MTCH2 have been linked to mitochondrial calcium handling. A method to rescue ESC cell interconversion in MTCH2 KO cells would greatly benefit the work. E.g the use of Drp1 downregulation in MTCH2 $-/-$ cells may distinguish between importance of active fusion per se or for an elongated network. Furthermore, the authors should show ESC exit from naive pluripotency by another method than solely Nanog expression and localization in Fig. 4.

Minor points:

The authors should include the complete Glc and Gln metabolic tracing data for MTCH2 $-/-$ 2i/L and AF cells (as for WT in Fig 3 B) at least in supplement.

MTCH2 was recently shown to regulate histone acetylation via differential localization of the pyruvate dehydrogenase complex. Does the same mechanism exist in ESC cells?

How can the authors exclude that MTCH2 regulation of apoptosis is not playing a role at all in ESC cell pluripotency?

The authors should state how many cells were used for the PA-GFP data in Fig,1 D.

Reviewer #4 (Remarks to the Author):

As requested, I focused my attention to quality of the metabolic analysis provided by Bahat et al. This is meant to demonstrate that in MTCH $-/-$ mESC cells, the defect in elongation/fusion of mitochondria is associated to a shift from pyruvate oxidation (fueled by glucose) to glutaminolysis. To support this claim, the authors performed a "kinetic flux measurement" using either ^{13}C -glucose or ^{13}C -glutamine. Unfortunately, there are some fundamental problems that need to be addressed to accept the claims.

Different from what they wrote, the authors didn't perform a kinetic experiment (e.g. as in PMID 16936719), but a single-point analysis. The results were interpreted as in the case of an end-point analysis, i.e. neglecting differences in metabolite pools. This simplification is only justified if the ^{13}C -labeling has reached the isotopic equilibrium. Depending on cell type, 6 hours incubation with ^{13}C might be insufficient to achieve stationarity of ^{13}C in the mitochondria. Did the authors verify that this essential prerequisite is met?

The changes in ^{13}C labeling shown in Figure 3b are marginal. The only number that is of relevance for the analysis is the labeling of citrate, that reports on the relative contribution of glutaminolysis or pyruvate to oxidative TCA cycle. Apart of the statistical significance, the change is of max 5%, e.g. from 40 to about 45%. By no mean, this result indicates a "strong decrease in pyruvate oxidation". Firstly, the change is marginal, i.e. max 10% of the flux. Secondly, glutamine and glucose-derived carbons account only for 50% citrate. This means that about 50% comes from other unknown sources and through undefined routes. For example, the difference in ^{13}C -citrate may result from a difference in ^{13}C -pyruvate and not by a difference in pyruvate oxidation. Thirdly, it's all based on relative contributions. It could also be that the pyruvate oxidation remains constant and the difference are only caused by a slight increase in glutamine uptake.

Overall, the current analysis doesn't support the claim.

Point-by-point response to the referees' comments

Response to Reviewer #1

It has become clear that the glycolytic metabolism is distinct between different states of pluripotency in the recent years. The authors here identify a novel regulator for mitochondrial fusion, mitochondrial carrier homolog 2 (MTCH2), which is critical for this transition (naïve to primed) in mouse embryonic stem cells (mESCs). Cells lacking this marker failed to exit naïve pluripotency, while overexpression of the profusion marker mitofusin 2 (MFN2) rescued this phenotype. The authors build on their previous interesting work describing that a loss of mitochondrial dynamics results in embryonic lethality during development, suggesting that these embryos failed to properly switch their metabolic dynamics towards a more primed state of pluripotency. By using the naïve to primed interconversion, the authors concurrently provide valuable insights on the role of mitochondrial metabolism during early development.

The authors apply a whole array of different techniques to support and validate their findings, adding to the comprehensive nature of their research and strengthening the scientific value of their conclusions.

The authors primarily validated the effects of MTCH2 knock-out (KO) in mouse embryonic fibroblasts (MEFs). They clearly show a rescue of MTCH^{-/-} with MTCH-GFP, however it would be interesting to see a comparison between MTCH^{-/-}+MTCH-GFP and MTCHF/F. They further show that expression of MFN2, a critical regulator of mitochondrial fusion, reduced mitochondrial fragmentation, which definitively proves that the cause of this fragmentation is a lower fusion rate. As with the MTCH-GFP comparison, a comparison with MTCHF/F may serve to further confirm their findings.

We thank the reviewer for this important comment. The experiments presented in Figure 1A, 1C and 1E were performed at the same time, and thus the results presented for *MTCH2^{F/F}* MEFs in Figure 1A are the reference for the cells presented in Figure 1C and 1E. We added dashed lines above the bars in Figures 1C and 1E to mark the percentages of the different mitochondria morphologies scored for the *MTCH2^{F/F}* MEFs (presented in Fig. 1A), and added to the text the % rescue of cells with elongated mitochondrial network in each case (MTCH2-GFP rescues 84.6% and MFN2-Myc rescues 86.1% as compared to *MTCH2^{F/F}*). The new additions in the text appear on p. 4, lines 60-61 and 67-68, respectively, in the revised manuscript.

The authors use a series of elegant experiments to show that MTCH2 KO maintains mESCs in a naïve state of pluripotency. They firstly show that they could derive MTCH2^{-/-} ESCs in 2i LIF conditions, albeit at a lower derivation efficiency than expected. Furthermore, when comparing MTCH2^{F/F} to MTCH2 KO with Cre recombinase, the authors observe slower cell growth, however interestingly the MTCH2 KO cells preserved naïve pluripotency. It would add to the scientific strength of the paper if the authors could speculate on the cause of the observed reduction in derivation efficiency or growth.

We thank the reviewer for this insightful comment. We hypothesize that the slower growth of the *MTCH2^{F/F}* ESCs in culture is due to the decrease in mitochondrial function (shown in Fig. 2G, H).

Slower growth for the *MTCH2*^{F/F} ESCs was also detected in the generation of teratomas *in vivo* (Fig. 2C). Based on these findings, we would speculate that the reduced derivation efficiency of *MTCH2*^{-/-} ESCs from blastocytes is also due to slower growth, since blastocytes derived at E3.5 need to proliferate at a high rate in culture to form successful colonies that will establish an ESC line. We have now added a sentence to the revised manuscript stating this point: “The above results are consistent with the idea that *MTCH2* deletion in ESCs results in mitochondria hyper-fragmentation and decreased mitochondrial function, which lead to slower growth of ESCs and to a decrease in E3.5 derivation efficiency.” (p. 6, lines 102-104).

The authors were able to further couple the effects of mitochondrial fusion to previously reported destabilization of mitochondrial DNA and mitochondrial respiratory function. They show that *MTCH2* regulates mitochondrial dynamics required for the exit from naïve pluripotency by comparing the priming of *MTCH2*^{F/F} cells and *MTCH2*^{-/-} towards EpiLCs. The authors observed a dampening in metabolic shift from pyruvate oxidation to a more primed metabolism, mitochondrial glutaminolysis. It is interesting that mitochondrial pyruvate oxidation is reduced in primed EpiLCs, while it has been previously shown that primed EpiSCs have a higher glycolytic rate compared to mESCs (Gu et al., Cell Stem Cell. 2016, Zhou et al. EMBO J. 2012), resulting in higher levels of pyruvate in the cytoplasm.

We again thank the reviewer for this important comment. We have originally interpreted our results as a decrease in pyruvate oxidation in primed EpiLCs, however we have now noticed (thanks to a comment by Rev #4) that labeling of citrate, that reports on the relative contribution of pyruvate to oxidative TCA cycle, was only marginally changed after priming (Fig. 3B, left panel). To assess whether the change in labeled citrate resulted from a difference in pyruvate oxidation, we used uniformly labeled pyruvate, and found that the labeling of citrate directly from pyruvate only marginally changed after priming (New Fig. S3A in the revised manuscript). These results suggest that after priming there is no difference in pyruvate oxidation but rather a decrease in glucose utilization for replenishing TCA cycle metabolites. Moreover, there is an increase in glutamine utilization for replenishing TCA cycle metabolites (Fig. 3B, right panel), suggesting that following priming, the levels of TCA-cycle metabolites are mainly maintained by an increase in glutamine utilization. This new paragraph appears on p. 7, lines 127-132 in the revised manuscript. These new results better fit with the literature mentioned by this reviewer (Gu et al., Cell Stem Cell. 2016, Zhou et al. EMBO J. 2012).

The authors were also able to further relate the inability of *MTCH2*^{-/-} EpiLCs to properly shift their metabolism to mitochondrial glutaminolysis to high levels of H3 acetylation. This successfully demonstrates not only that the cells exhibited hallmarks of naïve pluripotency, but also that *MTCH2* is involved in the regulation of epigenetic mechanisms.

Their transcriptomic analysis serves as a thorough validation of the effect of *MTCH2* on pluripotency, contributing a useful dataset to the stem cell community. The addition of qPCR and immunostaining constitutes an appropriate validation of the RNASeq data.

Finally, the authors adequately validate mitochondrial elongation as an early driving force of the naïve to primed pluripotency interconversion by correlating the effects of MFN2-Myc overexpression to mitochondrial morphology, NANOG and OCT4 expression levels. Taken together the authors effectively show that a shift in mitochondrial morphology is sufficient for reprogramming of mitochondrial metabolism, which in turn leads to epigenetic and

transcriptomic changes consistent with the naïve to primed interconversion.

Overall sentence build could be improved, for greater clarity and readability. There are many spelling errors, even in the abstract. The authors should address these.

We thank the reviewer for all the comments that appear here and below. We gave our paper to an English expert, whom corrected all the errors.

Lines 49-53, good background overview, but a bit cumbersome for the reader

We have shortened these two sentences for greater clarity and readability (p. 3, lines 41-43 in the revised manuscript).

Lines 121-123 repetitive

We understand this comment however we refer here for the first time in detail to the established interplay between mitochondria metabolism and the epigenome (Acetyl-CoA and histone acetylation). Thus, it is important to keep this sentence.

Line 136-138 needs to be clarified in more detail, to clearly show that there are two groups: MTCH2^{-/-} and MTCH2^{F/F}, which both have ESCs and EpiLCs. It was difficult to immediately understand this without referring to the figure.

We rephrased this part as follows: "The analysis was performed on four groups of cells: MTCH2^{F/F} cells grown in either 2i/L (ESCs) or in AF (EpiLCs) conditions, and MTCH2^{-/-} cells grown in either 2i/L (ESCs) or in AF (EpiLCs) conditions." (p. 8, lines 153-155 in the revised manuscript).

Response to Reviewer #2

The manuscript from Atan Gross and colleagues describes the effects of genetic inactivation of MTCH2 in MEFs and pluripotent cells. MTCH2 regulates mitochondria fusion in MEFs, while its absence in ES cells leads to fragmentation and reduced biogenesis and respiration. During the initial 48 hours of exit from naive pluripotency mitochondria start to elongate and shift their metabolism from Glucose to Glutamine consumption. All these effects are mitigated in the absence of MTCH2. Moreover, the transcriptional changes associated with exit from naive pluripotency are in large part abrogated in MTCH2 null ES cells.

Experiments are well performed, statistical analyses seems appropriate, and results are in large part novel, while some confirm previous studies.

A few points should be addressed to strengthen authors' conclusions.

1) no direct measurement of fusion in ES cells has been performed. The authors should measure GFP spreading as in figure 1D. This would support the claim that MTCH2 directly regulates fusion in ES cells, as indicated in the title and abstract of the manuscript.

We thank the reviewer for this important comment. To address this point, we performed a mitochondria fusion assay using ESCs prepared from genetically engineered mice with photo-activatable/convertible mitochondria (PhAM; green-to-red (New Ref #25)) crossed to *MTCH2^{F/F}* mice, and confirmed that *MTCH2^{-/-}* mitochondria exhibit a lower fusion rate (slower decrease in red fluorescence protein (RFP) intensity; New Fig. S2A in the revised manuscript). These new results are described on p. 5, lines 85-89 in the revised manuscript.

2) the flux analysis presented in figure 3B shows that in *MATCH2* fl/fl after 48h in AF there is a drastic reduction in some metabolites (e.g. Fumarate, Malate and Glutamine) labelled by ¹³C-Glucose.

Panel C shows the AF to 2i/L ratio in *MTCH2* fl/fl and *-/-* cells.

Somehow the two panels do not appear to be consistent, because in 3C the AF to 2i/L ratio in *MATCH2* fl/fl is between 0.6 and 0.8 for Fumarate, Malate and Glutamine, indicating only a mild reduction in labelling of such metabolites.

To avoid confusion I would suggest to present both datasets in the same way, plotting the fraction of metabolites labelled by ¹³C-Glc/Gln, as in Figure 3B.

We thank the reviewer for this important comment and apologize for the confusion. To address this point, we now present both data sets in the same way, plotting the fraction of metabolites labelled by ¹³C-Glc/Gln for the *MTCH2^{F/F}* cells (Fig. 3B) and for the *MTCH2^{-/-}* (New Fig. S3B in the revised manuscript). In addition, in panel C we corrected the ratios to be consistent with the data presented in Fig. 3B and New Fig. S3B. The modified text describing these results appears on p. 7, lines 135-138 in the revised manuscript.

3) the results indicating increased levels of Histone Acetylation are interesting, yet there is not a clear causal link with the metabolic changes observed.

Is Acetyl-CoA more abundant in *MTCH2^{-/-}* ES cells?

We thank the reviewer for this insightful comment. To address this point, we measured the levels of Acetyl-CoA in whole cell lysates using LC-MS (all four sample sets: *MTCH2^{F/F}* naïve ESCs and *MTCH2^{F/F}* primed EpiLCs, *MTCH2^{-/-}* naïve ESCs and *MTCH2^{-/-}* primed EpiLCs). We did not find any differences between the samples (New Fig. S3C in the revised manuscript). These new results are described on p. 8, lines 147-150. We have also added a sentence to the Discussion stating that "...further studies are required to elucidate the exact mechanism by which *MTCH2* regulates...the metabolic-epigenetic link." (p. 11, lines 236-237 in the revised manuscript).

4) In figure 4 the authors claim that expression of MFN2 is sufficient to drive exit from naive pluripotency. Such conclusions are based on the downregulation of *Nanog*.

Given that *Nanog* appear dispensable for maintenance/induction of naive pluripotency, the authors should support their conclusions by looking at other naive markers (e.g. *Klf4*, *Esrrb*, *Tfcp2l1*) and early differentiation markers (e.g. *Fgf5*, *Otx2*).

We thank the reviewer for this very important comment. To address this point, we looked at additional naïve markers (*ESRRβ* and *Klf4*) and early differentiation markers (*T-brachyury* and *FOXA2*). As in the case of *Nanog*, we found that *MTCH2^{-/-}* naïve ESCs expressing *MFN2-Myc* showed repression of nuclear estrogen related receptor beta (*ESRRβ*) expression levels (New Fig.

4B in the revised manuscript; see images in the left panels and bar graph in the top right panel). It should be emphasized that ESRR β is a downstream target of Nanog and a key regulator of mouse naïve pluripotency (New Ref #27), and thus these results strengthen our initial results with Nanog. Importantly, in cells expressing MFN2-Myc there was a correlation between the increase in elongated mitochondria (increase in aspect ratio of mitochondria) and the decrease in ESRR β expression levels (decrease in nuclear ESRR β intensity; New Fig. 4B, bottom right panel). The new text describing these results appears on p. 9, lines 180-186 in the revised manuscript.

Importantly, expression of MFN2-Myc in *MTCH2*^{F/F} naïve ESCs, maintained in naïve 2i/L conditions, also repressed nuclear ESRR β expression levels (New Fig. 4D in the revised manuscript). In the *MTCH2*^{F/F} naïve ESCs expressing MFN2-Myc there was also a correlation between the increase in elongated mitochondria and the decrease in ESRR β expression levels (New Fig. 4D, bottom right panel). The new text describing these results appears on p.10, lines 196-200.

Interestingly, we found that *MTCH2*^{-/-} and *MTCH2*^{F/F} naïve ESCs expressing MFN2-Myc did not show repression of a third naïve marker, nuclear Kruppel-like factor 4 (Klf4; data not shown). The new text describing these results appears on p.9, lines 186-189, and on p.10, lines 200-202, respectively. We also checked the expression levels of lineage differentiation markers (T-brachyury and FOXA2) in *MTCH2*^{F/F} and *MTCH2*^{-/-} naïve ESCs expressing MFN2-Myc/elongated mitochondria and downregulating Nanog expression, and could not detect expression of these markers (data not shown). The new text describing these results appears on p.10, lines 207-211 in the revised manuscript. In addition, we tested the expression of two post-implantation markers, Fgf5 (cat # sc-7914; Santa Cruz) and Otx2 (cat # AF1979; R&D systems), suggested by the reviewer, but were unsuccessful in detecting a proper signal after priming.

Collectively, our studies reveal that enforced elongation of mitochondria in naïve ESCs, even without eliminating the 2i/L conditions, is sufficient to trigger partial exit from naïve pluripotency (since Nanog and ESRR β are repressed but Klf4 is not) and without entering differentiation. These findings possibly relate to the recently described “Formative Pluripotency Phase”, an intermediate phase proposed to exist as part of a developmental continuum between the naïve and primed pluripotent states, marked by the downregulation of only part of the essential transcription factor program that maintains naïve pluripotency (New Ref #29, 30). Moreover, induction of lineage specific markers (e.g., T-brachyury and FOXA2) was not evident in the formative phase, suggesting that terminating the naïve circuitry precedes the expression of lineage markers (New Ref #31). This new discussion paragraph appears on p.11, lines 221-231 in the revised manuscript.

5) minor comment: line 80-81: rephrase, lethality by E7.5 cannot be due to defects in ESCs, rather due to defects already present at E3.5, which could be studied in ESCs.

Thank you and we rephrased this sentence accordingly (p. 4, lines 71-73).

6)minor comment: a previous work from the Gross laboratory characterised MTCH2 in ESCs as a regulator of apoptosis via tBID. Is such role completely unrelated from the effects observed in the current manuscript? The authors should discuss the two studies and the possible implications.

We thank the reviewer for this insightful comment related to our previous publication. In our

previous publication (Ref #22) we have demonstrated that the presence of MTCH2 sensitizes mitochondria to the addition of recombinant tBID to permeabilized ESCs, however we did not explore whether the endogenous BID-MTCH2 pathway plays a physiologically relevant role in apoptosis of ESCs. To examine whether loss of MTCH2 resulted in changes in the basal levels of apoptosis, we measured the levels of apoptosis in *MTCH2^{F/F}* and *MTCH2^{-/-}* naïve ESCs using propidium iodide (PI) and annexin V staining (both by FACS) and did not find any differences (New Fig S1A in the revised manuscript). This new data is described on p. 4, line 76.

Response to Reviewer #3

The mitochondrial protein MTCH2 has been previously described to regulate mitochondrial functions as well as apoptosis and metabolism. MTCH2 was also found to be an important player in the regulation of hematopoietic stem cell fate by the authors of this manuscript. Here, they have revealed that MTCH2 is also required for naïve-to-primed interconversion of murine ESCs and have linked this to its novel role as a promoter of mitochondrial fusion. This is of relevance to the growing interest in mitochondrial dynamics and metabolism as key regulators of stem cell fate. Their characterization of *MTCH2^{-/-}* ESCs reveals interesting differences in mitochondrial metabolism between WT and *MTCH2^{-/-}*, particularly when studying the shift between naïve to primed states. Interestingly, overexpression of the mitochondrial fusion factor MFN2 appears to be sufficient to promote ESC reprogramming. However, the underlying mechanism is unclear and therefore the argument that normal regulation of mitochondrial metabolism and histone acetylation is downstream of MTCH2-dependent mitochondrial fusion remains largely correlative and the underlying mechanism remains to be defined.

Major points:

1. How does MTCH2 affect mitochondrial fusion? The data in Fig 1 convincingly shows that *MTCH2* KO results in a fragmented morphology. The results of a PA-GFP fusion-experiment combined with the partial rescue of mitochondrial morphology in *MFN2^{-/-}* cells overexpressing *MTCH2* suggests that it may play a pro-fusion role. It should be noted that the mitochondrial morphology changes are far less pronounced in ESC WT vs *MTCH2^{-/-}* cells. The authors previously showed in LSK cells (Maryanovich et al., 2015) that *MTCH2* depletion causes mitochondrial enlargement due to reduced levels of mitochondrial Drp1. How can these discrepancies be explained? Drp1 phosphorylation status could be looked at along with a blot showing how all major mitochondrial dynamics factors are affected by *MTCH2*. Without further mechanistic insight it remains a possibility that mitochondrial dynamics regulation by *MTCH2* is an indirect effect of metabolic changes. Similarly, how can the authors exclude that *MTCH2* regulation of apoptosis is not playing a role at all in ESC cell pluripotency?

We thank the reviewer for this important comment. In our previous paper (Ref #10) we indeed reported that *MTCH2*-deficient LSK cells show less mitochondrial localization of DRP1, and that these findings correlated with the increase in mitochondrial size and volume. In addition, in the previous paper we speculated that the increase in mitochondrial size/volume was due to hyperfusion of mitochondria, however the findings in the current manuscript clearly indicate that the mitochondrial morphology aberrations in the *MTCH2^{-/-}* cells are largely due to a defect in

mitochondrial fusion.

In an attempt to obtain insights into the molecular basis of these mitochondria morphology changes in *MTCH2*^{-/-} ESCs, we monitored the expression levels of the major mitochondrial dynamics regulators in *MTCH2*^{F/F} and *MTCH2*^{-/-} naïve ESCs. Western blot analysis of whole cell lysates indicated that there is no significant differences in the expression levels of most of these proteins between the *MTCH2*^{F/F} and *MTCH2*^{-/-} cells (New Fig. S2B in the revised manuscript). Notably, there was a decrease in MFN1 and OPA1 levels in all three *MTCH2*^{-/-} clones, but this decrease was accompanied by a decrease in TOM20, which may indicate a reduction in mitochondria mass.

We also assessed the mitochondrial (heavy membrane) and cytosolic levels of DRP1 and of its phosphorylated forms (p-DRP1 S637) by subcellular fractionations followed by Western blot analysis. In both cases, we did not detect significant differences between the *MTCH2*^{F/F} and *MTCH2*^{-/-} naïve ESC samples (New Fig. S2C in the revised manuscript). We also attempted to assess the mitochondrial and cytosolic levels of the other phosphorylated form of DRP1, p-DRP1 S616, using the antibody from Cell Signaling (Cat #3455), but could not detect a band at the correct size. All these new data were described on p. 5, lines 90-97 in the revised manuscript. We have also added a sentence to the Discussion stating that "...further studies are required to elucidate the exact mechanism by which MTCH2 regulates mitochondria fusion/elongation..." (p. 11, lines 236-237).

To address the possibility that MTCH2 regulation of apoptosis is playing a role in determining ESC pluripotency, we measured the basal levels of apoptosis in *MTCH2*^{F/F} and *MTCH2*^{-/-} naïve ESCs using propidium iodide (PI) and annexin V staining (both by FACS). In these experiments, we did not detect differences in the levels of apoptosis between the *MTCH2*^{F/F} and *MTCH2*^{-/-} naïve ESCs (New Fig S1A in the revised manuscript), suggesting that the role of MTCH2 in apoptosis does not seem to be related to the effects observed in the current manuscript. This new data is described on p. 4, line 76.

2. Another discrepancy exists in the regulation of OXPHOS by MTCH2. In murine hemopoietic stem cells and skeletal muscle, MTCH2 loss increases OXPHOS. This should be discussed. It is also unclear how significant the OXPHOS changes are with regards to the MTCH2 ^{-/-} ESC cell fate.

We thank the reviewer for raising this important point. Indeed, we previously reported that mitochondria respiration/function increases upon MTCH2 deletion in hematopoietic stem cells (HSCs) and skeletal muscle cells (Ref #10, 11). We hypothesize that loss of MTCH2 in these cells initially results in a decrease in mitochondria respiration/function (as seen in ESCs), and the increase observed is a compensatory mechanism to correct for the defect and keep the cells alive. In the case of ESCs, we hypothesize that increasing mitochondria respiration/function may not be sufficient to enable normal embryonic development, and thus the mitochondrial defect is not corrected and the embryos eventually die at E7.5. This new discussion paragraph appears on p. 6, lines 105-111.

3. In Fig. 4, MFN2 overexpression is used to drive mitochondrial fusion and to argue that this helps drive ESC naïve-primed conversion. This is a key message of the manuscript and therefore

other methods to substantiate these data are important. How certain can the authors be that alternative functions of MFN2 are not in play. For example, both MFN2 and MTCH2 have been linked to mitochondrial calcium handling. A method to rescue ESC cell interconversion in MTCH2 KO cells would greatly benefit the work. E.g the use of Drp1 downregulation in MTCH2 $-/-$ cells may distinguish between importance of active fusion per se or for an elongated network. Furthermore, the authors should show ESC exit from naïve pluripotency by another method than solely Nanog expression and localization in Fig. 4.

We thank the reviewer for these very insightful comments.

To address the 2nd point, to show ESC exit from naïve pluripotency by another method than solely Nanog expression, we looked at additional naïve markers (ESRR β and Klf4) and early differentiation markers (T-brachyury and FOXA2). As in the case of Nanog, we found that *MTCH2*^{-/-} naïve ESCs expressing MFN2-Myc showed repression of nuclear estrogen related receptor beta (ESRR β) expression levels (New Fig. 4B in the revised manuscript; see images in the left panels and bar graph in the top right panel). It should be emphasized that ESRR β is a downstream target of Nanog and a key regulator of mouse naïve pluripotency (New Ref #27), and thus these results strengthen our initial results with Nanog. Importantly, in cells expressing MFN2-Myc there was a correlation between the increase in elongated mitochondria (increase in aspect ratio of mitochondria) and the decrease in ESRR β expression levels (decrease in nuclear ESRR β intensity; New Fig. 4B, bottom right panel). The new text describing these results appears on p. 9, lines 180-186 in the revised manuscript.

Importantly, expression of MFN2-Myc in *MTCH2*^{F/F} naïve ESCs, maintained in naïve 2i/L conditions, also repressed nuclear ESRR β expression levels (New Fig. 4D in the revised manuscript). In the *MTCH2*^{F/F} naïve ESCs expressing MFN2-Myc there was also a correlation between the increase in elongated mitochondria and the decrease in ESRR β expression levels (New Fig. 4D, bottom right panel). The new text describing these results appears on p.10, lines 196-200.

Interestingly, we found that *MTCH2*^{-/-} and *MTCH2*^{F/F} naïve ESCs expressing MFN2-Myc did not show repression of a third naïve marker, nuclear Kruppel-like factor 4 (Klf4; data not shown). The new text describing these results appears on p. 9, lines 186-189, and on p.10, lines 200-202, respectively. We also checked the expression levels of lineage differentiation markers (T-brachyury and FOXA2) in *MTCH2*^{F/F} and *MTCH2*^{-/-} naïve ESCs expressing MFN2-Myc/elongated mitochondria and downregulating Nanog expression, and could not detect expression of these markers (data not shown). The new text describing these results appears on p.10, lines 207-211 in the revised manuscript. In addition, we tested the expression of two post-implantation markers, Fgf5 (cat # sc-7914; Santa Cruz) and Otx2 (cat # AF1979; R&D systems) but were unsuccessful in detecting a proper signal after priming.

Collectively, our studies reveal that enforced elongation of mitochondria in naïve ESCs, even without eliminating the 2i/L conditions, is sufficient to trigger partial exit from naïve pluripotency (since Nanog and ESRR β are repressed but Klf4 is not) and without entering differentiation. These findings possibly relate to the recently described “Formative Pluripotency Phase”, an intermediate phase proposed to exist as part of a developmental continuum between the naïve and primed pluripotent states, marked by the downregulation of only part of the essential transcription factor

program that maintains naïve pluripotency (New Ref #29, 30). Moreover, induction of lineage specific markers (e.g., T-brachyury and FOXA2) was not evident in the formative phase, suggesting that terminating the naïve circuitry precedes the expression of lineage markers (New Ref #31). This new discussion paragraph appears on p.11, lines 221-231 in the revised manuscript.

To address the 1st point and distinguish the importance of active fusion per se from an elongated network for the exit from naïve pluripotency, we tested whether prompting the elongation of mitochondria by expression of a dominant negative form of the pro-fission protein dynamin-related protein 1 (DRP1-dn-YFP) would also induce exit from naïve pluripotency. We found that *MTCH2^{F/F}* and *MTCH2^{-/-}* naïve ESC expressing DRP1-dn-YFP showed elongated mitochondria and repression of both Nanog and ESRR β expression levels (New Fig. S4A-D in the revised manuscript; see images in the left panels and bar graphs in the right panels; note that DRP1-dn-YFP appears in green and is marked by white arrows in the top left images). Thus, active fusion per se is not required but rather an elongated network for the exit from naïve pluripotency. These new results are described in the abstract (p. 2, lines 25-28) and on p. 10-11, lines 212-220 in the revised manuscript.

Minor points:

The authors should include the complete Glc and Gln metabolic tracing data for *MTCH2^{-/-}* 2i/L and AF cells (as for WT in Fig 3 B) at least in supplement.

We thank the reviewer for this comment and we have included the data sets for *MTCH2^{-/-}* 2i/L and AF after labeling with ¹³C glucose and ¹³C glutamine (New Fig S3B, left and right panels, respectively).

MTCH2 was recently shown to regulate histone acetylation via differential localization of the pyruvate dehydrogenase complex. Does the same mechanism exist in ESC cells?

These findings were described by one of our collaborators but are unpublished yet (the manuscript describing these findings will probably be submitted soon). We still do not know if the same mechanism exists in ESCs.

How can the authors exclude that MTCH2 regulation of apoptosis is not playing a role at all in ESC cell pluripotency?

We thank the reviewer for this comment. To address the possibility that MTCH2 regulation of apoptosis is playing a role in determining ESC pluripotency, we measured the basal levels of apoptosis in *MTCH2^{F/F}* and *MTCH2^{-/-}* naïve ESCs using propidium iodide (PI) and annexin V staining (both by FACS). In these experiments, we did not detect differences in the levels of apoptosis between the *MTCH2^{F/F}* and *MTCH2^{-/-}* naïve ESCs (New Fig S1A in the revised manuscript), suggesting that the role of MTCH2 in apoptosis does not seem to be related to the effects observed in the current manuscript. This new data is described on p. 4, line 76.

The authors should state how many cells were used for the PA-GFP data in Fig,1 D.

Thank you. We used 30 cells to analyse each group for the PA-GFP data in Fig. 1D (this sentence was added to the legend).

Response to Reviewer #4

As requested, I focused my attention to quality of the metabolic analysis provided by Bahat et al. This is meant to demonstrate that in MTCH^{-/-} mESC cells, the defect in elongation/fusion of mitochondria is associated to a shift from pyruvate oxidation (fueled by glucose) to glutaminolysis. To support this claim, the authors performed a “kinetic flux measurement” using either ¹³C-glucose or ¹³C-glutamine. Unfortunately, there are some fundamental problems that need to be addressed to accept the claims.

Different from what they wrote, the authors didn't perform a kinetic experiment (e.g. as in PMID 16936719), but a single-point analysis. The results were interpreted as in the case of an end-point analysis, i.e. neglecting differences in metabolite pools. This simplification is only justified if the ¹³C-labeling has reached the isotopic equilibrium. Depending on cell type, 6 hours incubation with ¹³C might be insufficient to achieve stationarity of ¹³C in the mitochondria. Did the authors verify that this essential prerequisite is met?

We thank the reviewer for this important comment. A recent paper by Carey et al (Ref #1) reported that ¹³C-labeling has reached isotopic equilibrium after 4 hours with labeled glucose and glutamine in mouse ESCs. We have cited these findings in the revised manuscript, and corrected the procedure we performed from “kinetic flux measurement” to “We measured the changes in metabolites' carbon labeling following six hours of incubation with uniformly labeled ¹³C substrates (as it was shown that isotopic equilibrium is reached by this time point in mouse ESCs (Ref #1))...” (p. 7, lines 124-127 in the revised manuscript).

The changes in ¹³C labeling shown in Figure 3b are marginal. The only number that is of relevance for the analysis is the labeling of citrate, that reports on the relative contribution of glutaminolysis or pyruvate to oxidative TCA cycle. Apart of the statistical significance, the change is of max 5%, e.g. from 40 to about 45%. By no mean, this result indicates a “strong decrease in pyruvate oxidation”. Firstly, the change is marginal, i.e. max 10% of the flux.

Secondly, glutamine and glucose-derived carbons account only for 50% citrate. This means that about 50% comes from other unknown sources and through undefined routes. For example, the difference in ¹³C-citrate may result from a difference in ¹³C-pyruvate and not by a difference in pyruvate oxidation.

Thirdly, it's all based on relative contributions. It could also be that the pyruvate oxidation remains constant and the difference are only caused by a slight increase in glutamine uptake.

Overall, the current analysis doesn't support the claim.

We thank the reviewer for this very important comment. We completely agree that the labeling of citrate that reports on the relative contribution of pyruvate to oxidative TCA cycle was only marginally changed after priming. To assess whether the change in labeled citrate resulted from a difference in pyruvate oxidation per se, we used uniformly labeled pyruvate, and found no differences in citrate labeling (New Fig. S3A in the revised manuscript). These results suggest that after priming there is no difference in pyruvate oxidation but rather a decrease in glucose utilization and a preferential increase in glutamine utilization for generating TCA cycle metabolites. We have therefore modified the text accordingly: “Using gas-chromatography mass-spectrometry, we found that in *MTCH2^{F/F}* EpiLCs there is a decrease in glucose utilization and an increase in glutamine utilization for replenishing TCA-cycle metabolites (Fig. 3B, left and right panels, respectively). Notably, the labeling of citrate from glucose or directly from pyruvate only marginally changed after priming (Fig. 3B, left panel, and Fig. S3A, respectively), suggesting that following priming, the levels of TCA-cycle metabolites are mainly maintained by an increase in glutamine utilization. This new paragraph appears on p. 7, lines 127-132 in the revised manuscript. We have also corrected the Abstract to: “During this interconversion, wild-type ESCs elongate their mitochondria and alter their glucose and glutamine utilization for replenishing TCA-cycle metabolites.” (p. 2, lines 20-22).

REVIEWERS' COMMENTS:

Reviewer #1 (Remarks to the Author):

The authors have adequately addressed my comments, so I can recommend publication now.

Reviewer #2 (Remarks to the Author):

In the revised manuscript the Authors addressed the majority of points I raised. Authors' conclusions have been strengthened by the new results, therefore I support the publication of this interesting manuscript.

Reviewer #3 (Remarks to the Author):

The authors have carefully addressed my concerns on the original version of their manuscript and added a significant number of additional experiments to support their key findings.

They now provide evidence that it is the maintenance of the mitochondrial network rather than fusion per se that is required for the exit from naïve pluripotency. Thus, together with the finding that overexpression of MFN2 decreases Nanog expression, the authors provide now compelling evidence for a strict correlation between mitochondrial elongation and exit from naïve pluripotency.

Moreover, the authors clarified at least to some extent the underlying metabolic deficiency of Mtch2^{-/-} cells. Rather than reduced pyruvate oxidation, they provide now evidence that Mtch2 affects the metabolic switch from glucose to glutamine utilization that occurs during exit from pluripotency.

Together, the additional experiments further substantiate the central role of Mtch2 and mitochondrial elongation for the exit from pluripotency and, although some mechanistic aspects still need to be clarified, the present findings will be of broad interest in the field.

Reviewer #4 (Remarks to the Author):

The metabolic characterization in this manuscript has lost its traction, relevance, and significance. The authors have revisited the interpretation of the labeling experiments. Now, the results indicate that loss of MTCH2 induces a ~10% increase in glutamine uptake, while the glucose contribution remains constant. An independent validation of this result is missing. For instance, uptake rates could have been measured directly by mass spectrometry or (better) radioactive counting. However, the same conclusion was drawn by Carey et al.

Even though the original claim on a "strong decrease in pyruvate oxidation" was removed, there is still a tendency to over-interpret the metabolic results. For instance, terms such as "switch" should be removed in the case of such a marginal difference.

Abstract: "...ESCs elongate their mitochondria and alter their glucose and glutamine utilization for replenishing TCA-cycle metabolites." should be corrected to "...ESCs elongate their mitochondria and SLIGHTLY alter their glutamine utilization"

Page 7: "... there is a decrease in glucose utilization and an increase in glutamine utilization" should be corrected to "... there is a 10% decrease in relative glucose utilization and a corresponding increase in relative glutamine utilization". Bear in mind that the two contributions don't sum to 100% and, thus, the change in labeling could have a different explanation.

Page 7: "the levels of TCA-cycle metabolites are mainly maintained by an increase in glutamine utilization." This sentence is misleading, because glutamine maintains TCA cycle intermediates also in naïve mESCs.

Page 7: "we found that the switch from glucose to glutamine utilization" should be corrected to "we found that the increase in relative glutamine utilization": in naïve cells, less than 30% of the M+2 citrate originates from glucose (Fig S3a). Hence, the TCA does not depend on glucose, not even in the baseline naïve state.

Reviewer #4 (Remarks to the Author):

The metabolic characterization in this manuscript has lost its traction, relevance, and significance. The authors have revisited the interpretation of the labeling experiments. Now, the results indicate that loss of MTCH2 induces a ~10% increase in glutamine uptake, while the glucose contribution remains constant. An independent validation of this result is missing. For instance, uptake rates could have been measured directly by mass spectrometry or (better) radioactive counting. However, the same conclusion was drawn by Carey et al.

Even though the original claim on a “strong decrease in pyruvate oxidation” was removed, there is still a tendency to over-interpret the metabolic results. For instance, terms such as “switch” should be removed in the case of such a marginal difference.

Abstract: “...ESCs elongate their mitochondria and alter their glucose and glutamine utilization for replenishing TCA-cycle metabolites.” should be corrected to “...ESCs elongate their mitochondria and SLIGHTLY alter their glutamine utilization”

Corrected

Page 7: “... there is a decrease in glucose utilization and an increase in glutamine utilization” should be corrected to “... there is a 10% decrease in relative glucose utilization and a corresponding increase in relative glutamine utilization”. Bear in mind that the two contributions don't sum to 100% and, thus, the change in labeling could have a different explanation.

Corrected

Page 7: “the levels of TCA-cycle metabolites are mainly maintained by an increase in glutamine utilization.” This sentence is misleading, because glutamine maintains TCA cycle intermediates also in naïve mESCs.

This sentence was removed

Page 7: “we found that the switch from glucose to glutamine utilization” should be corrected to “we found that the increase in relative glutamine utilization”: in naïve cells, less than 30% of the M+2 citrate originates from glucose (Fig S3a). Hence, the TCA does not depend on glucose, not even in the baseline naïve state.

Corrected